# Evaluation of Thermal Changes of the Sole Surface in Horses with Palmar Foot Pain: A Pilot Study

**DOI:** 10.3390/biology12030423

**Published:** 2023-03-10

**Authors:** Cristian Zaha, Larisa Schuszler, Roxana Dascalu, Paula Nistor, Tiana Florea, Kálmán Imre, Ciprian Rujescu, Bogdan Sicoe, Cornel Igna

**Affiliations:** 1Surgery Clinic, Faculty of Veterinary Medicine, University of Life Sciences “King Michael I”, 300645 Timisoara, Romania; 2Dermatology Department, Faculty of Veterinary Medicine, University of Life Sciences “King Mihai I”, 300645 Timisoara, Romania; 3Department of Animal Production and Veterinary Public Health, Faculty of Veterinary Medicine, University of Life Sciences “King Michael I”, 300645 Timisoara, Romania; 4Management and Rural Development Department, Faculty of Management and Rural Tourism, University of Life Sciences “King Michael I”, 300645 Timisoara, Romania; 5Diagnostic Imaging, Faculty of Veterinary Medicine, University of Life Sciences “King Michael I”, 300645 Timisoara, Romania

**Keywords:** lameness, palmar foot pain, thermography

## Abstract

**Simple Summary:**

A pilot study was conducted to investigate the accuracy of thermography used to detect changes in the local temperature of horse feet and to compare the thermal patterns observed on the sole surface after training. The study group included eight horses with unilateral forelimb palmar foot pain (*n* = 8) and healthy contralateral limbs (*n* = 8). Four additional non-lame horses served as controls. We compared the local temperature of the frog and toe areas of the limbs affected with palmar foot pain in the study group with the other non-lame limbs in the study group and control group. The temperatures of the two selected areas were considered when determining differences between the three groups of horse limbs. The temperature of the frog area did not increase, and its area did not expand after training in the limbs affected with palmar foot pain. The temperature of the sole and its area both increased after training among the affected limbs of horses with palmar foot pain compared to the non-lame limbs. Based on the results obtained in this pilot study, we suggest a thermographic scanning of the toe and frog area to detect changes in the local temperature of the sole surface after training to discriminate the horses with palmar foot pain from non-lame ones.

**Abstract:**

Background: Horses with palmar foot pain do not show a typical increase in temperature in the palmar aspect of the hoof and heel due to low blood flow. The objectives of the current study were to determine the changes and differences in the thermographic pattern of the sole surface in horses with unilateral palmar foot pain and non-lame horses before and after training. We hypothesized that the thermal pattern and the local temperature of the frog area and toe area would be similar, with an increased local temperature after training in both lame and non-lame horses. A pilot study was conducted to investigate the accuracy of thermography used to detect changes in local temperature and to compare the thermal patterns observed on the sole surface after training. Methods: The study group included eight horses with unilateral forelimb palmar foot pain (*n* = 8) and healthy contralateral limbs (*n* = 8). Four additional non-lame horses served as controls. The horses were selected for the study based on the following criteria: forelimb with unilateral chronic progressive lameness and positive reactions when performing the hoof test and frog wedge test, degenerative findings of the navicular bone, and improvement in lameness after perineural analgesia of the medial and lateral palmar digital nerve. The local temperatures of the frog and toe areas were measured before and after training in the affected and contralateral limbs in the study group and both forelimbs in the control group using an FLIR E50 thermal camera. Receiver operating characteristic (ROC) curve analysis was applied to check the sensitivity and specificity of the results for the studied groups. Results: The thermal patterns of the hoof presented differences between the three groups of horse limbs. After training, the temperature of the sole surface increased, and its area increased in the limbs affected with palmar foot pain compared with the non-lame limbs in the study group and the limbs in the control group. The temperature of the frog area did not increase after training in the limbs affected with palmar foot pain compared with the same area in the other groups. The ROC curve analysis demonstrated the medical applicability of this tool and showed that thermography is a reliable diagnostic test to correctly discriminate between non-lame horses and those with palmar foot pain. Conclusions: We recommend thermographic scanning of the toe and frog area to detect changes in the local temperature of the sole surface after training to discriminate the horses with palmar foot pain from non-lame ones. Further investigation is required to clarify whether the observed thermographic imaging features of the sole surface are characteristic for horses with palmar foot pain.

## 1. Introduction

Palmar foot pain represents a common cause of lameness in horses and can be the result of navicular disease, navicular bursitis, desmitis to the distal sesamoidean impar ligament (DSIL), desmitis of the collateral ligaments of the distal interphalangeal joint (DIP joint), tendinitis of the deep digital flexor tendon, laminar tearing or bruising on the palmar region, distal phalanx damage, subsolar seromas, and abscesses [1,2,3]. Horses aged between 6 and 10 years are most commonly affected [4]. The etiology of palmar foot pain remains unknown, but two main theories, abnormal biomechanical stresses, and vascular compromise, have been suggested [5,6].

Diagnoses of palmar foot pain are based on history, physical examination, lameness examination, hoof testing, and perineural and/or intra-articular diagnostic anesthesia. Several imaging techniques (e.g., radiography (Rx), ultrasonography (US), nuclear scintigraphy (SCI), thermography (IRT), computed tomography (CT), magnetic resonance imaging (MRI)) are subsequently performed to ensure accurate diagnosis and proper treatment [7,8,9]. Turner [10] stated that in order to exacerbate the pressure on the navicular apparatus, a wooden pad should be placed under the frog area and the animal forced to land on it; this test is known as the frog wedge test.

Thermography is a non-invasive diagnostic method that involves scanning the surface temperature of an animal to find areas presenting potential vascular changes [11,12]. This ability to non-invasively assess vascular changes makes thermography an ideal imaging tool to aid in the diagnosis of certain lameness conditions in horses [13]. Thermography can support additional diagnostic information, together with Rx, US, CT, to detect vascular changes associated with hoof pain, such as laminitis, sole abscesses, navicular disease, and stress fracture, and, for the thoracolumbar region, it is used to scan the spinous processes of the thoracic vertebrae for diagnosis of inflammation and myositis of the region [14,15]. In thermographic imaging, the variation in the color pattern reflects thermal gradients; thus, the warmer areas with increased blood circulation are depicted as being white or red, while the cooler regions with insufficient blood supply appear blue or black [13]. Many studies emphasize the importance of thermography as a complement to ultrasonographic and radiographic examinations in the evaluation of soft tissue injuries and superficial bone lesions [12,14,15]. The highest body-surface temperatures (27–32 °C) are recorded in areas such as the flank, shoulders, head, and neck, while the lowest temperatures (24–26 °C) are recorded in the distal limbs [16]. The coronary band is 1 °C to 2 °C warmer than the remainder of the hoof [13].

Because thermography is non-invasive and inexpensive, it may be used to detect early changes prior to lameness occurring [17].

Turner et al. evaluated the local temperature of the caudal part of the foot to compare differences between non-lame horses and those suffering from navicular syndrome; horses with palmar foot pain did not display a typical increase in temperature in the palmar aspect of the hoof skin due to low blood flow after training [18].

The objectives of the current study were as follows:
To determine changes in the thermographic pattern of the sole surface in horses with unilateral palmar foot pain before and after training;To assess the differences in the thermal pattern of the sole in horses with palmar foot pain and non-lame horses before and after training;To verify that thermographic examination can be used to detect changes in the local temperature of the sole before and after training.

We hypothesized that the thermal pattern and the local temperature of the frog and toe area will not undergo temperature changes after training in the examined areas in any of the three groups.

## 2. Materials and Methods

### 2.1. Animal Selection

A prospective study was performed at the Surgery Clinic of the Faculty of Veterinary Medicine of Timisoara during the autumn season of 2021. Eight horses with unilateral forelimb lameness and four non-lame horses were selected. The horses’ feet were divided into the following three groups: group one (study group) represented by the affected forelimbs (*n* = 8), group two (study group) represented by the contralateral non-lame forelimbs (*n* = 8), and group three (control group) represented by the non-lame limbs of the non-lame horses (*n* = 4).

The inclusion criteria for the study group were as follows: more than three months of progressive unilateral forelimb lameness, minimum lameness grade of 2/10 on straight line, positive reactions to the hoof testers applied on the frog area and frog wedge pressure test, soundness after bilateral digital palmar perineural analgesia, radiographic observations, such as enthesophytes, on the medial and lateral aspects of the navicular bone, increased medullary sclerosis, and enlarged synovial invaginations [19,20]. No anti-inflammatory medication was administrated to the horses in the three weeks before the examination. The exclusion criteria for the study group were tenosynovitis and arthritis/arthrosis of the metacarpo-phalangeal joint.

The horses were clinically examined by two veterinarians with 10 and 14 years of experience, respectively, in musculoskeletal injuries, with a lameness exam performed according to the subjective European scale of lameness (0 is sound; 10 is non-weight-bearing lameness) [21]. The horses started to become lame while trotting on a hard surface, and this was even more obvious in a circle with the affected limb on the inside. The diameter of the circle around which the horses were lunged was 15 m. Lameness was obvious on the straight line in three horses and became worse during the lunge with the affected limb on the interior of the circle.

The hoof tester was applied on the frog area by squeezing the frog with one hand and the hoof wall with the other hand, to exert pressure on the navicular apparatus. The frog wedge test was performed by applying a wooden pad that was 10 cm long and 5 cm wide, with an angle of 15 degrees, under the palmar two thirds of the frog and forcing the horse to stand on this foot for 1 min by lifting the contralateral limb. The horses were walked after the frog test and a positive response was recorded if the lameness worsened.

To localize the lameness, perineural analgesia of the medial and lateral palmar digital nerve was performed in the pastern area at 1 cm proximal to the proximal edge of the collateral cartilage [17,22,23,24]. Before administration of the anesthetic solution, the hair was clipped, and the skin was scrubbed with povidone iodine. Perineural analgesia of the medial and lateral digital palmar nerve was performed using a 1.5-milliliter solution of 2% mepivacaine (Scandicaine^®^) for each nerve, and the size of the needle was 25 gauge and 16 mm [25]. Thirty minutes after perineural analgesia, the horses were subjected to a lameness examination [26].

Radiographic examination included the following views: lateromedial (LM), 65° dorsoproximal–palmarodistal oblique (DPr-PaDiO), and 45° palmaroproximal–palmarodistal oblique (PaPr-PaDIO) [19,20].

The control group was subjected to the same clinical exam, lameness and pressure tests, and radiographic imaging. For the control group, the lateral and medial digital nerve did not receive perineural analgesia. The exclusion criteria for the control group were no anti-inflammatory drugs and no history of tenosynovitis or hoof abscess in the last month.

### 2.2. Thermal Imaging and Data Recording

We evaluated the intensity of radiation in the infrared spectrum of the toe and frog area (Figure 1) before and after a 30-min lunging training session in each group. A qualitative assessment for the hoof wall area was performed before and after the training session. All the horses were subjected to the same intensity of exercise: 15-min walk on left and right hand and 15-min trot on left and right hand. The frog area was named El1, located in the middle third of the frog between the para-cuneal grooves at a size of 8 × 4 pixels. The toe area was named El2, located between the apex of the frog and hoof wall at a size of 6 × 4 pixels. The training area comprised a paddock built from quartz sand and the horses were lunged on both the right- and left-hand sides. The paddock surface was dry during the training sessions.

A thermographic exam was performed the day after the clinical exam and perineural analgesia.

All measurements for each horse in the study were conducted during the same season in the same environmental conditions, i.e., dry sand, air temperature of 18–22 °C, humidity between 60% and 70%, without extreme heat or cold weather. The acclimation period for thermal scanning was 30 min before training in a closed space with a temperature of 20–22 °C; after training, we did not include an acclimation period but proceeded directly with scanning [27]. The thermal images and measurements were taken by the same non-blinded operator for non-lame and lame horses at a 50 cm distance and 90° angle to the sole surface. Images were obtained using a FLIR E50 (FLIR Systems Inc., Wilsonville, OR, USA) infrared camera with the following parameters: 0.95 emissivity and 240 × 180 resolution for each image. The temperature range was between −20 and 650 °C, and the sensitivity was ≤0.05 °C. To identify the minimum and mean temperatures of the projection, we used FLIR Tools software for photo interpretation and the procedure was performed by the same operator.

For horses from the study group, the thermographic evaluation was performed between 10 a.m. and 12 p.m. on different days of the autumn season without differences in the outside temperature. For horses from the control group, the thermographic evaluation was performed in the same autumn season and on the same day so that the outside temperature did not influence the obtained results.

### 2.3. Statistical Analysis

The Kruskal–Wallis statistical test was applied to the temperature, and four procedures were used to evaluate each group. Comparisons were made between the control group, study group with the non-lame limb, and study group with palmar foot pain for the following four indicators (minimum temperature before training; minimum temperature after training; mean temperature before training; mean temperature after training) in both the toe and frog area. The Kruskal–Wallis statistical test was applied using IBM SPSS Statistics software version 23.0 (IBM, Armonk, NY, USA) with the statistical significance set at *p* < 0.05. Pairwise comparisons were also performed between the indicated groups using Dunn’s post hoc test with Bonferroni adjustment.

The receiver operating characteristic (ROC) curve is created from the sensitivity and specificity data and allows the creation of cut-off values. The purpose of ROC analysis is to identify the optimal threshold value to differentiate a positive from a negative result by obtaining the value of the area under the curve (AUC). To determine the ROC curve for toe area, the data were processed following attaching for each specimen of the group a score (temperature), respectively, which was a binary value (0—without lameness, 1—with lameness). To determine the ROC curve for frog area, we took into account the fact that in study group with palmar foot pain the temperature after training does not undergo obvious changes, the score attaching for each specimen was calculated as 1/t, where 1/t represent the inversion temperature for minimum and mean temperature before and after training. The PSPP software was used to obtain the results for ROC curve (PSPP – Analyze - ROC Curve). If the obtained AUC is greater than 0.9, then the result is excellent. If the obtained area is between 0.8 and 0.9, the result is very good; if it is between 0.7 and 0.8, the result is good; if it is between 0.6 and 0.7, the result is fair; and if it is less than 0.6, the result is dismissed.

The mean value for the minimum temperature prior to and subsequently after training was calculated separately for each group. The element of comparison of the evolution of the temperature after the training session between groups was the average value obtained from the mean of the minimum values of temperatures with the abbreviation of m.m.v.t.

## 3. Results

### 3.1. Animal Information

Out of fifty-eight horses examined during the autumn season of 2021 in the Surgery Clinic of Faculty of Veterinary Medicine from Timisoara, twelve were included in the study, and from these, four were sound horses and eight horses presented with unilateral forelimb lameness. All horses were Warmblood horses; three of the horses were used for cross-country sports and nine were used for leisure. The horses ranged in age from 6 to 14 years (mean 10.25 years and median 10 years). Among the horses from the study group, two presented with slight lameness on walking, and the other six had a history of lameness after exercise. Five horses were shod and three horses were bare-hoofed, and the horseshoes were removed from all horses before examination. After shoe removal, the hooves were trimmed and adjusted by a qualified farrier with more than 15 years of experience in horseshoe and hoof balance for racehorses. A control group composed of four barefoot, non-lame horses was also included in the study (*n* = 4). The horses from the control group were barefoot and a single forelimb was taken into account. These horses had private owners and were brought to the clinic for routine checks; they participated in the study with the consent of the owners. During the clinic examination, none of the horses presented with any musculoskeletal conditions.

### 3.2. Lameness Exam

Three horses were lame on the left forelimb and five were lame on the right forelimb; these ones represent the study group with palmar foot pain. All horses had a positive response to pain after flexion of the distal forelimb, which produced a transient exacerbation of the lameness. A lameness score of 4 was found in two of the studied horses, a score of 6 was found in three horses, and 7 in three horses, considering the European scale for lameness.

Six of the horses presented a mild reaction and two horses had a moderate reaction of retraction after applying the hoof tester on the frog area of the hoof. After using the wedge frog test, three of the horses presented a mild reaction, three horses a moderate reaction, and two horses an obvious reaction.

After perineural analgesia of the medial and lateral digital nerve, the lameness score was reduced to 0 in four horses (before: scores of 4 and 7), a score of 1 for three horses (before: scores of 6 and 4), and a score of 2 for two horses (before: scores of 6 and 7). After perineural analgesia, the response to the hoof tester on the palmar part of the hoof was negative in six horses, and two horses presented a mild reaction (Appendix A).

### 3.3. Radiographic Exam

In addition, 65° DPr-PaDiO radiographs of the hooves were taken for each of the horses included in the study group, revealing enthesophytes on the medial and lateral aspects of the navicular bone in three of the horses, along with increased medullary sclerosis of the navicular bone in four horses (Figure 2), while PaPr-PaDIO radiographs revealed sclerosis of the medullar cavity in two horses and increased opacity on the flexor border of the navicular bone for three horses. The LM revealed erosions of the flexor cortex of the navicular bone for two horses. Multiple lesions were observed in the case of six horses, and in only two horses did we observe a singular lesion [20]. The radiographic images were read by a veterinarian radiologist with 7 years of experience in the equine field.

The horses from the control group did not present any history of lameness and demonstrated no pain reaction after distal limb flexion. The clinical examination revealed no sign of lameness after walking, trotting, and lunging. The horses did not present any reaction to pain after applying the hoof tester and frog wedge test. The radiographic images did not reveal any changes in the structure of the navicular bone or abnormal opacity in the articular and flexor borders in the LM view. The 65° DPr-PaDiO radiographs of the hooves did not show any enthesophytes or arthrosis of the DIP, and the 45° PaPr-PaDIO radiographs did not detect sclerosis of the medullary cavity of the navicular bone.

### 3.4. Thermography Scan

#### 3.4.1. Thermography before Training

##### Thermography of the Toe Area

A small area of increased temperature was noticed between the frog and the hoof wall, which corresponds to the toe area, in the study group with palmar foot pain (Figure 3a and Table 1). The m.m.v.t. of the toe area was 20.6 °C.

A small area of increased temperature was present near the toe in the study group with the contralateral non-lame limb (Figure 4a and Table 1). The m.m.v.t. value of the toe area was 27.7 °C.

In the case of the control group, the toe area presented limited temperature increases (Figure 5a and Table 1). The m.m.v.t. for toe area was 24.0 °C.

##### Thermography of the Frog Area

In horses with palmar foot pain, the surrounding temperature was not raised in the frog area (Figure 3a and Table 1), and the m.m.v.t. in the frog area was 21.1 °C.

A small area of increased temperature was present near the frog area in the study group with the contralateral non-lame limb (Figure 4a and Table 1), with the m.m.v.t. in the frog area being 26.7 °C.

In the case of the control group, the frog area was characterized by a decrease in temperature (Figure 5a and Table 1), with the m.m.v.t. of the frog area being 22.8 °C.

#### 3.4.2. Thermography after Training

##### Thermography of Toe Area

A larger area of increased temperature was found in the toe region in horses with palmar foot pain (Figure 3b, Table 1 and Appendix A). The m.m.v.t. for the toe area was 27.7 °C.

In the study group with the contralateral non-lame limb, the local temperature of the toe area remained the same (Figure 4a, Table 1 and Appendix A). The m.m.v.t. for toe area was 28.8 °C.

In the control group the temperature rates in the toe area remained unchanged (Figure 5a, Table 1 and Appendix A), with m.m.v.t. of 26 °C.

The m.m.v.t. for the toe area reflected an obvious increase in temperature following training sessions, with values of up to 7.1 °C in the case of the palmar foot pain group, respectively, and lower rates for the non-lame limb group, with temperatures increased by only 1.1 °C. Similar to the latter, the temperature increased by only 2 °C in the case of the control group.

After training sessions, the thermal pattern seen in animals from the study group with palmar foot pain was made up of a wider area of increased temperature in the toe region, compared to the same area in the others non-lame groups, with *p* = 0.005.

##### Thermography of the Frog Area

In the study group with palmar foot pain, a smaller area of lowered temperature was found in the frog area (Figure 3b, Table 1 and Appendix A), with m.m.v.t of the frog at 22.2 °C.

In the study group with the contralateral non-lame limb, the area of increased temperature widened in the frog area (Figure 4b, Table 1 and Appendix A), with the m.m.v.t. being 28.7 °C.

In the control group, the frog area presented a wider area of increased temperature rates (Figure 5b, Table 1 and Appendix A), with the m.m.v.t. of the frog area at 25.2 °C.

The m.m.v.t. was low in the frog area for horses from the study group with palmar foot pain compared to values obtained for the horses from the contralateral non-lame limb and control groups, both before and after training sessions. The m.m.v.t. after training in horses with palmar foot pain increased by 1.1 °C compared to horses from the study group with non-lame limb, where a 2 °C temperature increase was recorded, and to horses from the control group, which showed an increase by 2.4 °C.

The thermal pattern of the frog area is characterized by decreased temperature rates in the case of the study group with the contralateral non-lame limb, compared to the same area in animals from the other non-lame horses, with *p* < 0.001.

##### Thermography of the Hoof Wall—Qualitative Assessment

In horses with palmar foot pain, before training along the hoof wall, there was a drop in temperature, and after the training, an increased area of high temperature can be seen along the hoof wall (Figure 3).

In the study group with the contralateral non-lame limb, before training, an area of decreased temperature was detected along the hoof wall compared with a widened area of increased temperature after training (Figure 4).

In the control group, an area of decreased temperature was detected along the hoof wall before training compared with a wider area of increased temperature detected after the training (Figure 5).

Before training, based on the thermal pattern, an area of increased temperature was detected along the hoof wall in the study group with the contralateral non-lame limb, compared to the same area for the horses with palmar foot pain and those from the control group.

After training, based on thermal pattern examination, the hoof wall temperature was increased over a wider area in the control group and in the study group with non-lame limbs, as compared to the study group with palmar foot pain.

### 3.5. Group Comparisons

The result obtained after applying the Kruskal–Wallis test show the *p* < 0.001 after comparison did not differ between the three groups (control group, study group non-lame limb, study group with palmar foot pain) successively following the four indicators for temperature for the toe and frog area. The exceptions for this comparison between the groups were that the minimum temperature for the toe after training had a significance level *p* = 0.005, and the mean temperature of the toe before training were the significance level was *p* = 0.01.

Pairwise comparisons led to significant differences for the frog area after training between the study group limbs with palmar foot pain and the control group (*p* < 0.001) (Figure 6) after Dunn’s post hoc test with Bonferroni adjustment. Pairwise comparisons led to a significant difference being found between the study group limbs with palmar foot pain and the study group non-lame limbs regarding the minimum and mean temperatures in the toe area before training (*p* < 0.001). Significant differences were also determined for the control group and study group non-lame limbs regarding the minimum temperature of the toe area after training (*p* = 0.002), with increased temperatures for the study group non-lame limbs. Significant differences were also determined for the mean temperature of the toe area after training between the control group and others study groups (*p* = 0.006) (Figure 7).

### 3.6. ROC Curve

The ROC curve for the inversion minimum temperature of the frog area before training was 0.97 and *p* < 0.05. The ROC curve for the inversion minimum temperature after training and for the mean temperature before and after training gave a value of the AUC of 0.98 and *p* < 0.05 (Figure 8a and Appendix A).

The ROC curve revealed, for the toe area, two results with an AUC of 0 for the minimum and mean temperature before training, one result with an AUC of 0.7 for the mean temperature of the toe after training, and one result with an AUC of 0.5 and *p* > 0.23 for the minimum temperature of the toe area before training (Figure 8b and Appendix A).

## 4. Discussion

Thermography of the sole before and after the training session allowed us to differentiate horses with palmar foot pain from non-lame ones; this is highlighted by the rejection of the null hypothesis regarding the minimum and mean temperature before and after training for the frog and toe area. The post-training thermal images from the control group, from the study group with palmar foot pain, and from the study group with non-lame limbs revealed that the local temperature and the thermal pattern of the sole undergo changes after training. The most significant changes in temperature were noticed in the frog and toe areas. The thermal pattern of the toe area after training indicates a significantly larger area of increased temperature in the toe region for the horses with palmar foot pain, as compared to the thermal pattern for the same area in non-lame horses. In addiiton, in horses with palmar foot pain, following thermal scanning after training, a smaller area of lowered temperature in the frog area was observed, as compared to the same area for non-lame horses.

Soroko and Howell report that in horses suffering from disorders affecting the navicular area, the animal is forced to land on the toe instead of the heel, resulting in stride-shortening. The thermographic examination revealed an intensification of blood flow in the sole area, supporting that these animals land improperly on their toes [28]. Our study rendered similar results for horses with palmar foot pain after training, with increased temperatures in the toe area, as to avoid pain in the palmar part of the hoof. The spot showing an increased temperature, identified using thermography, could be differentiated from the increased local temperature linked to the presence of an abscess by associating the pathology with the history of the animal and hoof radiography [29,30,31].

In our study, the differences in the minimum and mean temperature in the toe area between the control group, non-lame limbs study group, and limbs with palmar foot pain study group appeared because the horses with palmar foot pain were forced to land on the ground on the toe region while horses from the non-lame limbs group were forced to sustain their weight during movements. Thus, in the study group with palmar foot pain, the toe-area showing increased temperature became wider after training, comparable with the same area from the other groups. Buchner et al. indicates a compensation of the reduced loading of the lame limb by the contralateral non-lame limb [32]. This relationship is also visible during lameness, when reaction forces have a tendency to decrease in the lame limb and increase in the contralateral non-lame limb in order to compensate the dysfunction [33].

The pre-training temperature of the frog area in horses with palmar foot pain was decreased compared to the local temperature of the frog area in the other groups of horses. After training, the temperature in the frog area increased by 1.1 °C, on average, in horses with palmar foot pain. As for the other group of horses, the temperature increased by more than 2 °C. Turner [18] reports that in the case of horses with palmar foot pain, around 50% of them do not sustain an increase in temperature in the caudal foot because of low blood flow. This is in contrast with other inflammatory processes of the hoof, such as abscesses, bruises, and fractures, which are characterized by the presence of a “hot spot” at the site of injury, which increases in temperature in the case of exercise [13,18]. A warm spot indicates inflammation or increased circulation, while a cold spot reflects a reduction in blood supply, usually due to swelling, thrombosis, or scar tissue [29,30].

The temperature of the hoof wall increased in value after training for all three groups as a result of increased blood flow in the hooves with consecutive training [34]. Before training, a wider area of temperature was detected for the study group non-lame limbs compared with the same area from other groups as a result of compensatory vertical load bearing, which has the potential to lead to a secondary injury in non-lame horses [35].

All the horses from the studied group with palmar foot pain presented moderate or severe lameness after the clinical examination, a result that other authors have also revealed in their studies of navicular disease and palmar foot pain [6,8,24].

According to the ROC curve, which gave excellent results for three of the eight examined variables with an AUC of 0.98, one with AUC of 0.97, and one with AUC of 0.7, thermography provides excellent clinical utility to correctly identify the horses with palmar foot pain and non-lame ones. For three variables, the value of the AUC was less than 6 (two results AUC of 0, one result AUC of 0.5), so the result was dismissed. As a result, for AUC values, we recommended thermographic scanning of the toe and frog area to claim changes in temperature for horses with palmar foot pain. Compared to other diagnostic imaging procedures (US, Rx, CT, MRI), thermography is inexpensive, requires no harmful radiation, carries no risk of injury to the veterinarian, and does not require the horses to be sedated [11,12,17].

To obtain a successful thermal image the horses must have a dry coat and skin, without sweating, should not have performed physical therapy in the past 24 h, and no anesthetic protocol should be used [15,33,36]. The thermal scanning can be performed within 45 min of perineural anesthesia, without the risk of changes in the local temperature of the limb surface [37]. In our study, the thermography was performed on a dry sole, without artefacts, sweating, the next day after the lameness examination, perineural anesthesia, and without general sedation.

During exercise, working muscles produce substantial quantities of heat, which must be eliminated from the body by sweating, in order to prevent the animal from overheating [18]. In our study, sweating does not appear to influence the thermographic images due to a lack of intense exercises and due to the absence of the eccrine gland on the surface of the frog [38,39].

Our recommendation regarding the evaluation of the sole surface local temperature before and after training is to perform the thermography on a dry area, without artefacts, in an environment without extreme temperature variations and to perform the training session on a dry surface.

The obtained results indicate a disturbance of the local temperature in the toe region in horses with palmar foot pain, and we suggest that this increase in toe temperature may be related to differences in loading characteristics of the foot. Further investigations based on sole surface thermography are required. These should aim to: (1) assess the correlation between biomechanics changes and sole thermography in palmar foot pain horses; (2) compare the thermal pattern areas for non-lame and affected horses before and after training using specific software for area measurement in the SI units (International System of Units); (3) discover whether there is a difference in the thermal pattern in horses with bilateral palmar foot pain and those with unilateral palmar foot pain; and (4) identify the differences between horses with subclinical palmar foot pain and non-lame horses.

The limitations of the study were the small number of studied cases, the fact that horses with bilateral palmar foot pain were not studied, the moderate camera resolution, and the failure to determine the results on different support surfaces. The operator was not blinded during the marking of the study areas. Time elapsed after perineural analgesia was used to evaluate the lameness score, which could have obscured problems with many more structures than the palmar heel region alone. Another limitation is that we did not use MRI, CT, or SCI to properly diagnose lesions of the navicular bone, DDFT, navicular bursa, or collateral and impaired ligaments in the studied and control groups.

## 5. Conclusions

The minimum temperature in the toe area presented an obvious increase in temperature following training sessions in horses with unilateral palmar foot pain, compared with the same area in the other non-lame limbs. The local temperature in the frog area for horses with palmar foot pain is decreased compared to the temperature observed in the non-lame limbs, both before and after training.

Based on the results obtained in this pilot study, we suggest a thermographic scanning of the toe and frog area to detect changes in the local temperature of the sole surface after training to discriminate the horses with palmar foot pain from non-lame ones.

Further investigations, on a larger number of horses, are required to clarify that the observed thermal pattern can be proposed as being characteristic for horses with palmar foot pain.

## Figures and Tables

**Figure 1 biology-12-00423-f001:**
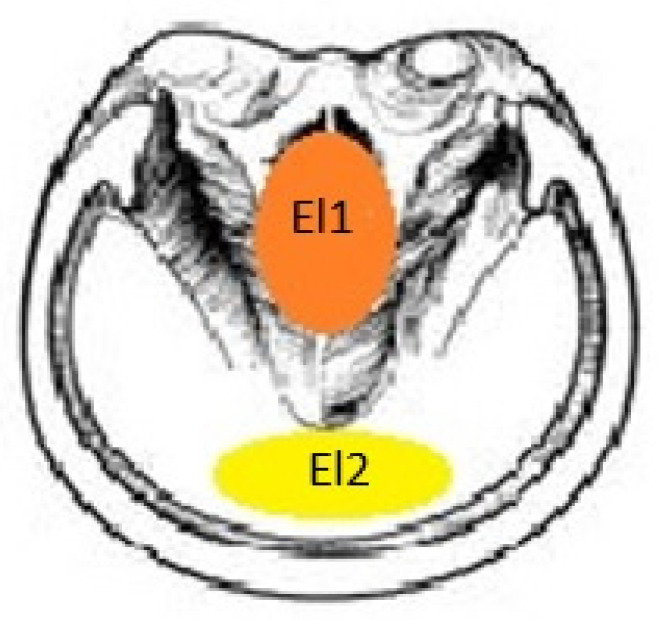
Sole surface: El1, frog area; El2, toe area.

**Figure 2 biology-12-00423-f002:**
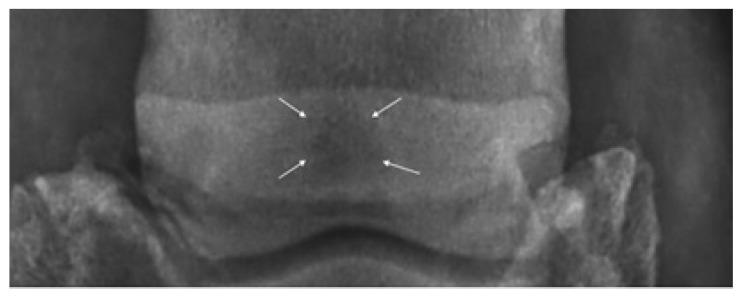
Horses with palmar foot pain: DPr-PaDiO radiolucent area (sclerosis of the medullary cavity) in the navicular bone (white arrows).

**Figure 3 biology-12-00423-f003:**
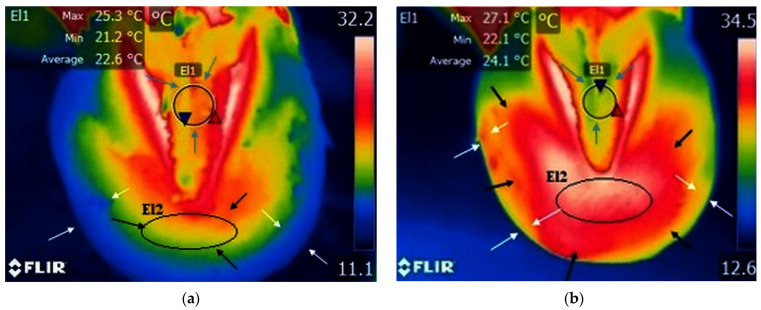
Study group, limbs with palmar foot pain: (**a**) sole surface before training—small surface of increased temperature in the frog area (blue arrows) and toe area (black arrows); reduced temperature along the hoof wall (white arrows); minimum temperature recorded in the area—blue triangle spot; maximum temperature recorded in the area—red triangle spot; (**b**) sole surface after training—increased temperature and wider surface of increased temperature in the toe area (black arrows); no obvious changes in local temperature in the frog area (blue arrows); increased temperature along the hoof wall (white arrows); minimum temperature recorded in the area—blue triangle spot; maximum temperature recorded in the area—red triangle spot.

**Figure 4 biology-12-00423-f004:**
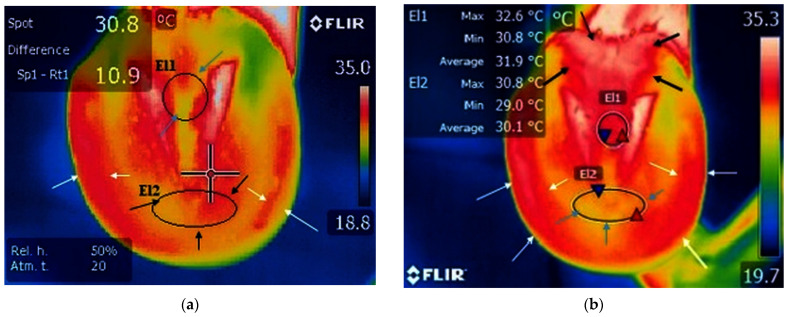
Study group, non-lame limbs: (**a**) sole surface before training—reduced temperature and small surface of temperature in the frog area (blue arrows); increased area of temperature on hoof wall surface (white arrows); reduced temperature in the toe area (black arrows); (**b**) sole surface after training—increased temperature and wider surface of increased temperature in the frog area (black arrows); increased surface of temperature along the hoof wall (white arrows); small surface of reduced temperature in toe area (blue arrows); minimum temperature recorded in the area—blue triangle spot; maximum temperature recorded in the area—red triangle spot.

**Figure 5 biology-12-00423-f005:**
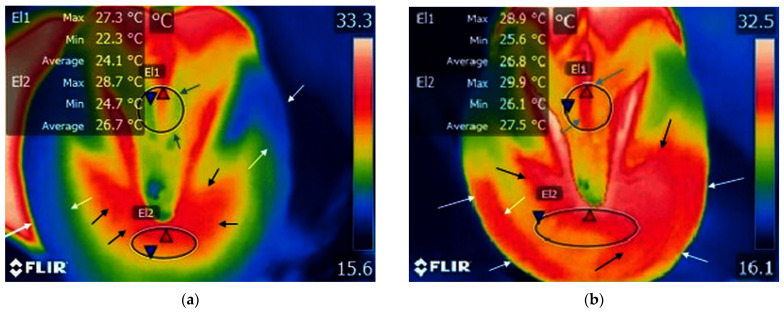
Control group: (**a**) small surface of increased temperature in the toe area (black arrows); small surface of increased temperature in the frog area (blue arrows); reduced temperature of the hoof wall (white arrows); minimum temperature recorded in the area—blue triangle spot; maximum temperature recorded in the area—red triangle spot; (**b**) increased local temperature and wider surface of increased temperature in the toe area (black arrows); increased local temperature and wider surface of increased temperature in the frog area (blue arrows); increased temperature along the hoof wall (white arrows); minimum temperature recorded in the area—blue triangle spot; maximum temperature recorded in the area—red triangle spot.

**Figure 6 biology-12-00423-f006:**
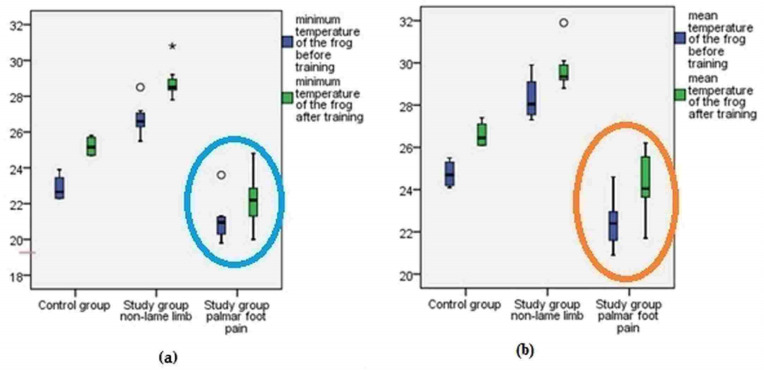
(**a**) Boxplot diagram showing the minimum temperature values of the frog area before and after training. Low values for minimum temperature before and after training are observed for the study group with palmar foot pain (blue circle). (**b**) Boxplot diagram showing the mean temperature values of the frog area before and after training. Low values for mean temperature before and after training for the study group with palmar foot pain are observed (orange circle); *—extreme outliers temperature, ᵒ—mild outliers temperature.

**Figure 7 biology-12-00423-f007:**
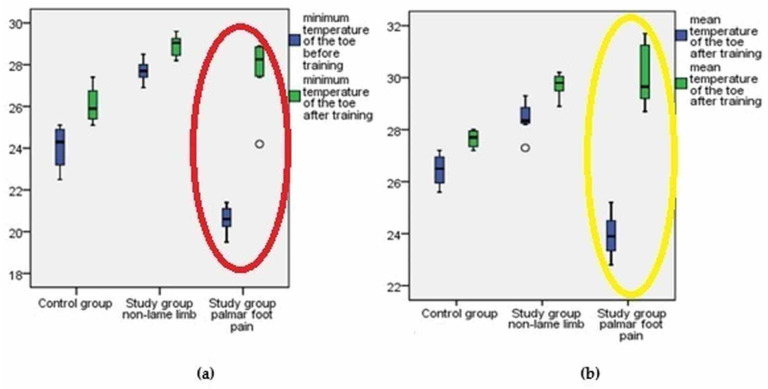
(**a**) Boxplot diagram showing the minimum temperature values of the toe area before and after training. An obvious difference is observed for the minimum temperature of the toe area before (blue mark) and after training (green mark) for the study group with palmar foot pain (red circle). (**b**) Boxplot diagram showing the mean temperature values of the toe area before and after training. Low values for mean temperature of the toe area are observed before (blue mark) and after training (green mark) for the study group with palmar foot pain (yellow circle); ᵒ—mild outliers temperature.

**Figure 8 biology-12-00423-f008:**
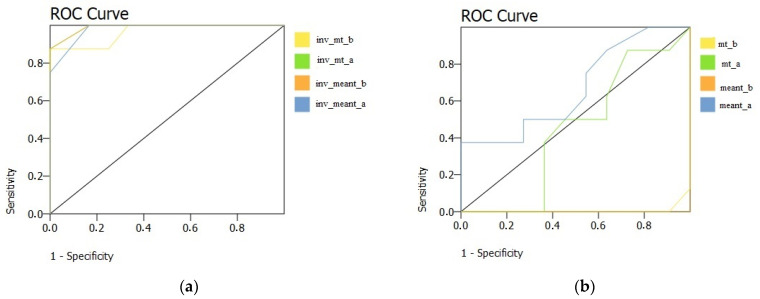
(**a**) Area under the curve (AUC) of the receiver operating characteristic (ROC) curve for frog surface showed the next values for the variants: inv_mt_b—inversion minimum temperature before training—value: 0.97; inv_mt_a—inversion minimum temperature after training—value: 0.98; inv_meant_b—inversion mean temperature before training—value: 0.98; inv_meant_a— inversion mean temperature after training—value: 0.98. (**b**) Area under the curve (AUC) of the receiver operating characteristic (ROC) curve for toe area showed differences between the variants: mt_b—minimum temperature before training—value 0; mt_a—minimum temperature after training value:0.4; meant_b—mean temperature before training—value: 0; meant_a—mean temperature after training value: 0.7.

**Table 1 biology-12-00423-t001:** Descriptive statistics for minimum, maximum, and mean temperature for each group.

				Minimum	Maximum	Median
Control group	frog area	minimum temperature	before	22.3	23.9	22.65
			after	24.1	25.5	24.70
		mean temperature	before	24.7	25.8	25.15
			after	26.1	27.4	26.45
	toe area	minimum temperature	before	22.5	25.1	24.30
			after	25.6	27.2	26.50
		mean temperature	before	25.1	27.4	25.90
			after	27.2	28.0	27.70
Study group, non-lame limb	frog area	minimum temperature	before	25.5	28.5	26.80
			after	27.8	30.8	28.40
		mean temperature	before	27.3	29.9	27.95
			after	28.8	31.9	29.35
	toe area	minimum temperature	before	26.9	28.5	27.55
			after	28.2	29.6	28.90
		mean temperature	before	27.3	29.3	28.35
			after	28.9	30.2	29.55
Study group, palmar foot pain limb	frog area	minimum temperature	before	19.8	23.6	21.20
			after	20.0	24.8	22.85
		mean temperature	before	20.9	24.6	22.95
			after	21.7	26.2	25.55
	toe area	minimum temperature	before	19.5	21.4	20.50
			after	24.2	28.9	27.70
		mean temperature	before	22.8	25.2	23.90
			after	28.7	31.7	29.30

## Data Availability

All the data obtained during the retrospective study are available in the Clinical Register of Large Animals from the Surgery Clinic.

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
