# Peer review of "Evaluation of Thermal Changes of the Sole Surface in Horses with Palmar Foot Pain: A Pilot Study"

_biology, 2023, doi:10.3390/biology12030423_

Round 1
Reviewer 1 Report (Previous Reviewer 2)
The manuscript needs to be edited by someone with a strong background in English writing.
Author Response
Dear reviewer,
The new versus of the manuscript was edited by a specialist from MDPI Language Editing Services

Reviewer 2 Report (Previous Reviewer 3)

Author Response
Dear reviewer,
Thank you for your attention to giving us important information to improve our manuscript. We are grateful to your insightful comments.

Round 2
Reviewer 1 Report (Previous Reviewer 2)
The authors have made some improvements to the manuscript but this manuscript does not give the reader enough information to use thermography as a diagnostic tool. Cut-off values would be very helpful to include from the ROOC curves and including some temperature measurements directly in the text would also be very useful. Please see further comments below:
Abstract:
ROC curves are created using sensitivity and specificity. It is unclear how you determined these numbers. I would assume based on your knowledge of the horses with palmar foot pain. However, if you know which group your thermographic images belong, you would have perfect sensitivity and specificity.
Lines 55-58: This is a very small group of horses to be able to claim this. You would need to test this on a large number of horses to validate this claim. This may be the next step
Introduction:
Lines 86-87: Thermography does not detect these pathologies. It may detect temperature changes that can be found with these pathologies. It may support taking additional diagnostic imaging of a specific region (i.e., radiographs, ultrasound, etc.)
Lines 103-104: Wouldn’t a low blood flow show a decrease in temperature?
Lines 112-115: While I get the hypothesis, it seems that it could be reworded so that your reader does not have to read it multiple times. I think what you are saying is that there will not be a difference in any of the three groups in thermal pattern or local frog temperature before or after training? Please consider rewording to make this more concise.
Materials and Methods:
Animal selection: I assume that this was a prospective study (horses were enrolled as they were seen). If this is the case, the horse information should be included in the results (i.e., you did not know which specific horses would be included when you started the study).
This section needs some improved organization. There are 2 places were inclusion/exclusion criteria are contained (lines 131-132, 138-144, 149-150), lameness is discussed twice (lines 125, 145-151).
Lines 118-122: These first 2 sentences say the same thing. Consider rewording the first sentence and deleting the “unilateral palmar foot pain of the forelimbs”.
Line 125: What is the grading scale for this lameness scoring (0-10?). Is there a reference that describes the grading? Please include this.
Line 134: For the control horses, I assume that these were clients of the hospital. It would be nice to have something included about how these horses were recruited (maybe they were client horses that were seen for routine care by the hospital?)
Line 140: Since these horses were lame on one forelimb, why would you do bilateral PD blocks? By bilateral do you really mean that you blocked both the medial and lateral nerves?
Line 167: 30 minutes is a very long time to wait for a horse to block. Allowing this much time may allow for migration of the local anesthetic proximally. It would be difficult to know if the horse blocked to a PD or it was blocking to something higher.
Statistical analysis: The Kruskal-Wallis test is for non-parametric data. Did you examine your data for normality? Because of the small sample size, a non-parametric test would make sense as the data may not be normal, but this sentence should be changed to indicate that the data was not normally distributed. Were comparisons made between groups (i.e., frog area of lame vs control horses)? Based on how this is written, it appears that statistical analysis only looked at differences within groups (before and after exercise). If there were comparison amongst groups, this needs to be stated.
Lines 216-217: What was your gold standard? To calculate sensitivity and specificity, you need a gold standard.
Results:
Lines 222-223: Please indicate that this was the study horses that had the lameness.
Thermography scans: It would be nice to see actual values of temperatures presented (thermography gave you objective data so it would be good to include it). If you are going to state that a temperature is increased, it would also be good to say what is increased by comparison (i.e., the frog was warmer than the tor region). By how much is the region increased? It is also unclear if the changes that you are reporting were significant. Please include.
Lines 262-263: I assume that this sentence pertains to the toe area.
Lines 263-264: This sentence contradicts what was stated in the first sentence of the paragraph (was the frog temperature increased after training or not). Please include how much the temperature increased (a median of ___ degrees).
Lines 271-273: This sentence is repetitive for the frog compared to an earlier sentence.
Table 1: This table would be more useful if it were easier to compare groups if the groups were columns (i.e., lame, non-lame control as column headers and frog area/toe area as columns). The Range values do not make sense. A range is reported as a minimum and maximum (23.3 – 26.1).
Lines 334-345: Is this comparison looking at the difference in the temperature change (before exercise and after exercise) among the 3 groups or is this looking at comparing the 3 groups before exercise and then comparing them again after exercise. How it is written is unclear.
Lines 346-357: I assume that these pairwise comparisons are related to the previous paragraph(s). All of this information should be in 1 paragraph (not multiple paragraphs).
Lines 358-360: When you mention smaller temperatures, do you mean the ROI or the degrees? If you mean the temperature itself, it would be better to use the word “decreased”. It would also be nice to have an objective value to include (what was the difference in temperature or size of the area between groups).
Lines 361-362: Was this a significant difference? Please state if this is the case.
Table 2: Chi-squared values should not be included in the Table. Please include actual values (median/range) for the 3 groups. I assume that this is the same information as Table 1. These Tables should be combined. P-values can be included in Table 1 and then this Table can be deleted.
Lines 366-375: The definition of AUC should not be included in the Results section. This is discussion material. The ROC results are appropriate for the results section.
Discussion
The discussion still needs to be reworked. It is wandering and not well organized. While this data appears to be useful, there does not appear to be enough information that your reader could take this tool and use it in a clinical case (which I assume would be the point of the paper). Specifically, what are the cut-off values for the ROC curve to differentiate a palmar foot pain foot from a control foot? How do you know that a horse with some other type of lameness would also not show this pattern? It seems like the next step would be to demonstrate that palmar hoof pain horses show a different pattern than horses with another source of lameness.
Please reword the first paragraph to be more succinct about your findings. It is good to state that you rejected your hypothesis but also state what you found that was the most interesting. The sentence in lines 401-405 does not state what you found – what is a particular pattern? If your reader is going to use thermography, what would they expect to find in a palmar foot pain horse vs a normal horse? Specifically what area(s) did you find the most helpful? It seems like lines 406-407 may be the big take away.
Lines 407-411: It is not appropriate for you to state your results and then combine this with a reference. This reference may lend support to what you found but you did not look at “vascular foramina” in your study.
Line 411-412: This sentence needs a reference
Lines 418-420: How do you know the horses “were forced to land on the ground in the toe region”? Your temperature pattern may indicate a toe first landing is most likely but unless you documented this pattern (which was not mentioned in the manuscript), you cannot state this.
Lines 441-445: These sentences just state facts. They need to be included in relevant areas of the discussion not just as blanket statements.
Lines 449-450: Which examined variables for the ROC curve would be most useful?
Author Response
Dear Reviewer,
We thank to you for the constructive and informative comments that significantly improved our revised manuscript.
Our study with pilot study with thermography of the sole surface in horses with unilateral palmar foot pain need further research on a large group of horses to check if the date present a normal distribution. At this moment the Kruskal-Wallis test for non-parametric data show differences after comparison the three groups of horses by took into consideration four indicators of temperature for frog and toe area before and after training. Your observation in comments regarding the ROC curve gave us a new perspective on temperature assessment which is presented in the next paragraphs.
We introduce new date in the Table (minimum, maximum for the variants and remove the range). The p value obtain from KW test was inserted into the paragraphs.
We calculate median for temperatures differences both for minimum and mean value for the three studied groups and include them between parentheses as additional data.
Below are the answers provided by us in a point-by-ponit shape.
ROC curves are created using sensitivity and specificity. It is unclear how you determined these numbers. I would assume based on your knowledge of the horses with palmar foot pain. However, if you know which group your thermographic images belong, you would have perfect sensitivity and specificity.
Response: As a result of the observations made, we reviewed the ROC curve and made changes. The PSPP test was used to obtain the results. To determine the ROC curve for toe area (Fig 1.), the data were processed following attaching for each specimen of the group a score (temperature), respectively a binary value (0 - without lameness, 1 - with lameness). To determine the ROC curve for frog area we took into account the fact that the temperature in study group with palmar foot pain after training does not undergo obvious changes, the score attaching for each specimen was calculated as 1/t, where t represent the minimum and mean temperature before and after training (Fig 2). Four result with AUC greater than 9, one result with AUC 7, one result AUC 5 and two result AUC 0 was obtained.
Fig 1. ROC curve for toe area: mt_b—minimum temperature before training; mt_a—minimum temperature after training; meant_b—mean temperature before training; meant _a—mean temperature after training.
Fig 2. Area under the curve (AUC) of the receiver operating characteristic (ROC) curve for frog surface showed a value of 1 for all the variants: inv_mt_b—reversal minimum temperature before training; inv_mt_a—reversal minimum temperature after training; inv_meant_b—reversal mean temperature before training; inv_meant_a— reversal mean temperature after training.
Lines 55-58: This is a very small group of horses to be able to claim this. You would need to test this on a large number of horses to validate this claim. This may be the next step
Response: we use the study as a pilot one to describe the use thermography in detection of local changes in horses with palmar foot pain. Further investigation is required on a large number of horses to confirm that the observed thermal pattern can be proposed characteristic for horses with palmar foot pain.
Introduction:
Lines 86-87: Thermography does not detect these pathologies. It may detect temperature changes that can be found with these pathologies. It may support taking additional diagnostic imaging of a specific region (i.e., radiographs, ultrasound, etc.)
Response, line 99-101: Thermography can support additional diagnostic information, together with Rx, US, CT, to detect vascular changes associated with hoof pain, such as laminitis, sole abscesses, navicular disease and stress fracture, and, for the thoracolumbar region, it is used to scan the spinous processes of the thoracic vertebrae for diagnosis of inflammationnd myositis of the region.
Lines 103-104: Wouldn’t a low blood flow show a decrease in temperature?
Response: Yes, a low blood flow decrease the local temperature, also, the autor ( Turner ) say: “horses with palmar foot pain did not display a typical increase in temperature in the palmar aspect of the hoof skin due to low blood flow after training.”The author say that local circulation is disturb in navicular disease. (ref 13: Turner, T.A. Diagnostic Thermography. Vet Clin of North America Equine Pract, 2001, 17(1):95-105. )
Lines 112-115: While I get the hypothesis, it seems that it could be reworded so that your reader does not have to read it multiple times. I think what you are saying is that there will not be a difference in any of the three groups in thermal pattern or local frog temperature before or after training? Please consider rewording to make this more concise.
Response, lines 138-143: The new version reaches the desired hypothesis on the subject. “We hypothesized that the thermal pattern and the local temperature of the frog and toe area will not undergo temperature changes after training in the examined areas in any of the three groups.“
Materials and Methods:
Animal selection: I assume that this was a prospective study (horses were enrolled as they were seen). If this is the case, the horse information should be included in the results (i.e., you did not know which specific horses would be included when you started the study).
Response, lines 146-164: Correct statement, we did not know the breed of the horse from the beginning. In the paragraph we add information about the breed, age, history of lameness and made changes in text in the direction of those recommended.
“A prospective study that included eight clinical cases thus three of the horse were used for cross-country sports and five were used for leisure and all of them present unilateral palmar foot pain of the forelimb. Two horses presented slight lameness on walking, and the other six had a history of lameness after exercise. The horses’ ages ranged from 6 to 14 years, and all of them were warmblood horses and were examined in the Surgery Clinic of the Faculty of Veterinary Medicine of Timisoara during the autumn season of 2021.”
This section needs some improved organization. There are 2 places were inclusion/exclusion criteria are contained (lines 131-132, 138-144, 149-150), lameness is discussed twice (lines 125, 145-151).
Response, lines 171-179: We combined the inclusion criteria from 131-132, 138-144,149-150 in one paragraph “The inclusion criteria for the study group were as follows: more than three months of progressive unilateral forelimb lameness, minimum lameness grade of 2/10 on straight line, positive reactions to the hoof testers applied on the frog area and frog wedge pressure test, soundness after bilateral digital palmar perineural analgesia and radiographic observations such as enthesophytes on the medial and lateral aspects of the navicular bone, increased medullary sclerosis and enlarged synovial invaginations [25, 26]. No anti-inflammatory medication was administrated to the horses in the three weeks before the examination.”
We removed the lines 125 about the lameness, remain the paragraph, lines 183-192.
Lines 118-122: These first 2 sentences say the same thing. Consider rewording the first sentence and deleting the “unilateral palmar foot pain of the forelimbs”.
Response: We edit the beginning of the paragraph by combine the lines 118 and 122. “A prospective study that included eight clinical with affected forelimb (n=8) and the contralateral non-lame ones (n=8) who represent the study group”
Line 125: What is the grading scale for this lameness scoring (0-10?). Is there a reference that describes the grading? Please include this.
Response: The reference for lameness grading is at nr 19. (Ross, M.W.; Dyson, S.J. Diagnosis and Management of lameness in the Horse, 2nd ed.; Elsevier, St. Louis, Missouri, 2010, pp. 78-79)
Line 134: For the control horses, I assume that these were clients of the hospital. It would be nice to have something included about how these horses were recruited (maybe they were client horses that were seen for routine care by the hospital?)
Response, lines 169-171: The horses from the control group had private owners and were used for leisure. These horses were brought to the clinic for routine control and to participate in the study with the consent of the owners. During the clinic examen the horses didn’t presented a musculoskeletal conditions.
Line 140: Since these horses were lame on one forelimb, why would you do bilateral PD blocks? By bilateral do you really mean that you blocked both the medial and lateral nerves?
Response, lines 200-201: Only for the affected limb was performed perineural analgesia. By bilateral block we referred to the medial and lateral digital palmar nerve of the affected limb.
Line 167: 30 minutes is a very long time to wait for a horse to block. Allowing this much timenfu may allow for migration of the local anesthetic proximally. It would be difficult to know if the horse blocked to a PD or it was blocking to something higher.
Response: Bidwell et all (https://doi.org/10.2746/0425164044848154) recomand: “When using a palmar digital nerve block, it is important to perform lameness evaluations between 15 mins and 1 h to be sure of effective nerve blockade.” Biavaschi Silva et all say: " Taking into account the fact that the volume of the drug may interfere with the specificity of the block, by blocking unwanted structures when using higher volumes, is interesting to make the block with the smallest amount of anesthetic capable of producing analgesia." We choose half an hour and used a small quantity of the local anesthetic to take care that the solution not block something higher. We add as a limitation of the study at the line 614-616
Statistical analysis: The Kruskal-Wallis test is for non-parametric data. Did you examine your data for normality? Because of the small sample size, a non-parametric test would make sense as the data may not be normal, but this sentence should be changed to indicate that the data was not normally distributed. Were comparisons made between groups (i.e., frog area of lame vs control horses)? Based on how this is written, it appears that statistical analysis only looked at differences within groups (before and after exercise). If there were comparison amongst groups, this needs to be stated.
Response, line 252-258:Because the volume of data is low, it was inconclusive to determine whether it follows a normal distribution. Thus, the Kruskal-Wallis non-parametric statistical test was used. Comparison was made between groups ( control group, study group with non-lame limb and study group with palmar foot pain) following four indicators ( 1. Minimum temperature before training; 2. Minimum temperature after training; 3. Mean temperature before training; 4. Mean temperature after training) both for toe and frog area.
Lines 216-217: What was your gold standard? To calculate sensitivity and specificity, you need a gold standard.
Response: the cases with palmar foot pain took into study were diagnosed based on anamnesis, lameness and radiological examen. The clinician have to choose whether the measurement, if the results are deemed clinically relevant or not. It is the outcome of this evaluation process that does or does not justify the use of the term “characteristic thermal pattern”, not the measurement in itself. (Van Weeren, P. R., Pfau, T., Rhodin, M., Roepstorff, L., Serra Bragança, F., & Weishaupt, M. A. (2018). What is lameness and what (or who) is the gold standard to detect it. Equine Vet. J, 50, 549-551)
Results:
Lines 222-223: Please indicate that this was the study horses that had the lameness.
Response: Add at line 279.
Thermography scans: It would be nice to see actual values of temperatures presented (thermography gave you objective data so it would be good to include it). If you are going to state that a temperature is increased, it would also be good to say what is increased by comparison (i.e., the frog was warmer than the tor region). By how much is the region increased? It is also unclear if the changes that you are reporting were significant. Please include.
Lines 262-263: I assume that this sentence pertains to the toe area.
Response: After training, the temperature in toe region increased and the area of increased temperature became wider. Add at line 325
Lines 263-264: This sentence contradicts what was stated in the first sentence of the paragraph (was the frog temperature increased after training or not). Please include how much the temperature increased (a median of ___ degrees).
Response, lines 320-353: The temperature in frog area did not show obvious changes after training for the horses with palmar foot pain (minimum temperature +1°C, mean temperature +1,7°C) compared with the same area for the other groups (control group: minimum temperature +2,3 °C; mean temperature 1,9°C; contralateral non-lame limb:minimum temperature + 1,9°C; mean temperature +2,3°C.
Lines 271-273: This sentence is repetitive for the frog compared to an earlier sentence.
Response: We remove from text the repetitive sentence
Table 1: This table would be more useful if it were easier to compare groups if the groups were columns (i.e., lame, non-lame control as column headers and frog area/toe area as columns). The Range values do not make sense. A range is reported as a minimum and maximum (23.3 – 26.1).
Response: We took your advice into account and edited the table and erase the range values. The table contain the minimum, maximum, mean value for each group. Lines 370-404.
Lines 334-345: Is this comparison looking at the difference in the temperature change (before exercise and after exercise) among the 3 groups or is this looking at comparing the 3 groups before exercise and then comparing them again after exercise. How it is written is unclear.
Response, lines 252-256: Comparison was made between groups ( control group, study group with non-lame limb and study group with palmar foot pain) following four indicators ( 1. Minimum temperature before training; 2. Minimum temperature after training; 3. Mean temperature before training; 4. Mean temperature after training) both for toe and frog area.
Rewrite at 436-441
Lines 346-357: I assume that these pairwise comparisons are related to the previous paragraph(s). All of this information should be in 1 paragraph (not multiple paragraphs).
Response, line 442-459: reorganize in one paragraph for pairwise comparison for frog and toe areaThe Dunn’s post hoc test with Bonferroni adjustment show comparison between groups (Grup 1 vs Group 2, Group 2 vs Group 3, Group 3 vs Group 1).
Lines 358-360: When you mention smaller temperatures, do you mean the ROI or the degrees? If you mean the temperature itself, it would be better to use the word “decreased”. It would also be nice to have an objective value to include (what was the difference in temperature or size of the area between groups).
Response, lines 360-364: degree of temperature. In non-lame horses, the values for the minimum and the mean temperature of the frog area were higher before training (minimum temperature +2,3 °C; mean temperature 1,9°C) compared with the same values for the frog area in horses with palmar foot pain (minimum temperature: median of +1°C, mean temperature: median of +1,7°C).
Lines 361-362: Was this a significant difference? Please state if this is the case.
Response, line 464-467: Yes, An increased temperature was detected for the toe area after training for the horses with palmar foot pain (minimum temperature: median of + 7,1°C, mean temperature: median of + 6,3°C) compared with non-lame horses (minimum temperature: median of + 3,7°C; mean temperature: median of +1,3°C).
Table 2: Chi-squared values should not be included in the Table. Please include actual values (median/range) for the 3 groups. I assume that this is the same information as Table 1. These Tables should be combined. P-values can be included in Table 1 and then this Table can be deleted.
Response: We edited the Tabel 2 but we can’t combine the Table 1 and Table 2. The first tabel (Table 1) shows the descriptive statistic of the data for minimum, maximum, mean for each group and the second table (Table 2) point out the comparison that were made between groups ( control group, study group with non-lame limb and study group with palmar foot pain) took into account the minimum / mean of the temperature for the toe and frog area before and after training. The content of Table took we insert in the paragraph at line 436-441, and the Table 2 we erase.
Lines 366-375: The definition of AUC should not be included in the Results section. This is discussion material. The ROC results are appropriate for the results section.
Response: We remove the definition of AUC from 366-375 to 271-275. The ROC result were maintained at 475-478.
Discussion
The discussion still needs to be reworked. It is wandering and not well organized. While this data appears to be useful, there does not appear to be enough information that your reader could take this tool and use it in a clinical case (which I assume would be the point of the paper). Specifically, what are the cut-off values for the ROC curve to differentiate a palmar foot pain foot from a control foot? How do you know that a horse with some other type of lameness would also not show this pattern? It seems like the next step would be to demonstrate that palmar hoof pain horses show a different pattern than horses with another source of lameness.
Please reword the first paragraph to be more succinct about your findings. It is good to state that you rejected your hypothesis but also state what you found that was the most interesting. The sentence in lines 401-405 does not state what you found – what is a particular pattern? If your reader is going to use thermography, what would they expect to find in a palmar foot pain horse vs a normal horse? Specifically what area(s) did you find the most helpful? It seems like lines 406-407 may be the big take away.
Response, lines 502-520: We reword the first paragraph and include description for the characteristic thermal pattern and which are the important area that are took into consideration for changes in local temperature.
“The null hypothesis regarding the minimum and mean temperatures of the frog area and toe region before and after training was rejected in all situations for all groups in the study. The significance level was p<0.05; therefore, the research hypothesis that the groups would show similar thermal patterns and the same local temperature in the frog and toe area after training was rejected. The thermal images obtained after training from the control group, study group limbs with palmar foot pain and study group non-lame limbs revealed that local temperature and the thermal pattern of the sole undergoes changes after training. The most significant changes in temperature was for frog and toe area. Based on the changes in local temperature after training, a characteristic thermal pattern was identified for horses with palmar foot pain were the temperature and the area of the frog region slightly increase but the temperature and area of temperature for toe area increase comparative with the non-lame horses were the temperature increase and the area of temperature became wider after training.”
Lines 407-411: It is not appropriate for you to state your results and then combine this with a reference. This reference may lend support to what you found but you did not look at “vascular foramina” in your study.
Response, lines 520-526: We reorganize the paragraph and bring explanation for the changes in local temperature that others authors found.
“Soroko and Howell reports the fact that in horses with affection in navicular area the animal make support on the ground with cranial part of the hoof by reduced weight bearing in the palmar foot area [29]. In horses with palmar foot pain, around 50% of them do not sustain an increase in temperature in the caudal foot because of low blood flow [18]. This is in contrast with other inflammatory processes of the hoof, such as abscesses, bruises and fractures, which are characterized by the presence of a “hot spot” in the site of injury, which increases in temperature in the case of exercise [13]. A warm spot indicates inflammation or increased circulation , while a cold ?spot reflects a reduction in blood supply, usually due to swelling, thrombosis or scar tissue [30, 31].”
Line 411-412: This sentence needs a reference
Response, line 526: ref 18
Lines 418-420: How do you know the horses “were forced to land on the ground in the toe region”? Your temperature pattern may indicate a toe first landing is most likely but unless you documented this pattern (which was not mentioned in the manuscript), you cannot state this.
Response: To reduce the pain in the palmar region of the hoof. Soroko and Howell report this also (reference nr 29) line.
Lines 441-445: These sentences just state facts. They need to be included in relevant areas of the discussion not just as blanket statements.
Response: insert in the paragraph at line 529-532
Lines 449-450: Which examined variables for the ROC curve would be most useful?
Response, lines 589-599: The next variable was relevant, which get a value of AUC more than 0,7:
- for frog area: reversal minimum temperature before and after training, reversal mean temperature before and after training;
- for toe area: mean temperature after training

Round 3
Reviewer 1 Report (Previous Reviewer 2)
Thank you for making some of the requested changes. The results and discussion sections do not flow well (need better organization). Inclusion of subheadings in the results would help fix this.
Specific comments:
Materials and Methods
Animal Information – since the study was prospective, all horse information should be included in the results section.
Statistical analysis – Please define PSPP test.
Results – There needs to be some reorganization of this section. The addition of sub-headings may help improve readability.
Thermography scan: It does not make sense to talk about differences in pre and post training before you present the pre training results. How do you know that there is an “increased temperature of the frog area”. It would make sense to first present what patterns were seen pre-exercise in all 3 groups. Are there differences in groups? Then discuss patterns in post-exercise in all 3 groups. Are there differences? Then make statements on what differences occurred comparing pre and post exercise. Having sub-titles under the section heading of “Thermography scan” would help organization. For instance “Thermography Before Training”. Within these sub-sections, toe and frog area could be discussed separately.
Lines 436-441: These sentences do not make sense. Please reword.
Lines 442-467: This needs to be reworded. It should not be worded by discussing statistical tests of null hypotheses. Please present when there were differences among groups. A sub-heading (could be something like “ Group comparisons”).
Line 469: What is “reversal minimum temperature?
Discussion: This section still needs significant rewriting. The writing is poor in some sections (lines 522-523). The meaning of some sentences does not make sense (lines 526-528). For paragraph (lines 534-544), the writing does not make clear comparisons from what they found and key literature findings. There need to be references that support what they found (toe first landings on lame leg and changes to the contralateral limb based on compensatory overloading?). I am not sure why there needs to be a discussion of the shoe, when the shoes were removed on all horses (lines 552-555).
Lines 556-562: Including this paragraph with no reference to what was done in this study does not make sense. Horses after training had to be evaluated immediately (no 10-20 minute acclimatization) and horses might have sweated during exercise. How does this affect your findings?
Lines 591-595: List which variables were useful based on ROC. It is not important to include those less than 6 (0.6?)
Conclusions: Statistical tests should not be mentioned here. The conclusions should be the main take home message not rehash the entire discussion. Would you recommend exercising the horse and then comparing the patterns between front feet? It seems like examining the toe area is the key area. Please delete the last sentence (lines 633-634).
Author Response
Dear reviewer,
All authors thank for reading and give comments to the manuscript.
We reorganize the Thermographic scan section and present the result of thermography before and after training sessions in two different sub-sections and another section where we present the evolution of the temperature.
The new version of the manuscript has been revised by a native English speaker.
Materials and Methods
Animal Information – since the study was prospective, all horse information should be included in the results section.
Response: we add paragraph with horse information in the results section.
Statistical analysis – Please define PSPP test.
Response, Line 273: Was a misspelling. The correct term is PSPP software.
Results – There needs to be some reorganization of this section. The addition of sub-headings may help improve readability.
Response: Reorganize the section and add sub-heading and sub-sections. 3.1. Animal information, 3.2. Lameness examination, 3.3. Radiographic examination, 3.4. Thermography scan, 3.5. Group comparisons, 3.6. ROC curve
Thermography scan: It does not make sense to talk about differences in pre and post training before you present the pre training results. How do you know that there is an “increased temperature of the frog area”. It would make sense to first present what patterns were seen pre-exercise in all 3 groups. Are there differences in groups? Then discuss patterns in post-exercise in all 3 groups. Are there differences? Then make statements on what differences occurred comparing pre and post exercise. Having sub-titles under the section heading of “Thermography scan” would help organization. For instance “Thermography Before Training”. Within these sub-sections, toe and frog area could be discussed separately.
Response: We change the result section for thermography and present separately the result obtained before training (line 345-365) and after training sessions (367-412).
At line 382-391, we present differences in thermal pattern after training for each group.
At line 392-412, we present differences in thermal pattern between groups.
Lines 436-441: These sentences do not make sense. Please reword.
Response: lines 413-441: we present the evolution of the temperature in toe and frog area. We compare the mean of minimum temperature value (m.m.t.v.) before and after training session between groups.
Lines 442-467: This needs to be reworded. It should not be worded by discussing statistical tests of null hypotheses. Please present when there were differences among groups. A sub-heading (could be something like “ Group comparisons”).
Response: We add a new sub-heading (3.5. Groups comparisons) and remove the information about statistical test of null hypothesis. In the lines 545-590 we present the differences among groups.
Line 469: What is “reversal minimum temperature?
Response: we correct the term “reversal” with inversion. The “inversion minimum temperature” represent the score 1/t calculated for frog area. Information at Lines 260-273
Discussion: This section still needs significant rewriting. The writing is poor in some sections (lines 522-523). The meaning of some sentences does not make sense (lines 526-528). For paragraph (lines 534-544), the writing does not make clear comparisons from what they found and key literature findings. There need to be references that support what they found (toe first landings on lame leg and changes to the contralateral limb based on compensatory overloading?). I am not sure why there needs to be a discussion of the shoe, when the shoes were removed on all horses (lines 552-555).
Response: we change and correct the discussion section.
Lines 522-523: correct at 625-629.
Lines 526-528: correct at 651-653.
Lines 534-544: correct at lines 636-654
552-555: remove the discussion with shoe.
Lines 556-562: Including this paragraph with no reference to what was done in this study does not make sense. Horses after training had to be evaluated immediately (no 10-20 minute acclimatization) and horses might have sweated during exercise. How does this affect your findings?
Response: we add bibliography for the paragraph (line 750). The author from the referee recommended thermography after 10-20 minute acclimatization. Our thermography was performed immediately after training session (lines 236-237).
Authors from referee say that sweat will not influence the thermographic examen if the intensity of the exercise is low and due to the fact that in horses the eccrine gland are missing from the surface of the frog (Simon et all, Talukdar et all).
Lines 591-595: List which variables were useful based on ROC. It is not important to include those less than 6 (0.6?)
Response: We add the useful variables that provide clinical utility at lines 708-711. We add the variables less than 0.6 with information that these will not provide clinical utility at line 711-713.
Conclusions: Statistical tests should not be mentioned here. The conclusions should be the main take home message not rehash the entire discussion. Would you recommend exercising the horse and then comparing the patterns between front feet? It seems like examining the toe area is the key area. Please delete the last sentence (lines 633-634).
Response: we reorganize the conclusion section at lines 805-815.
We delete the sentence from 633-634

Round 4
Reviewer 1 Report (Previous Reviewer 2)
Line 194: I assume that 12 am should be pm.
Results section
Lines 283-372 and 435-451: It seems that the results are repeated multiple times (qualitative assessment first and then quantitative); results should be reported once. The whole thermography results section needs to be organized and made more concise. It needs to be clear throughout when statistical comparisons were made (include P values). Please correct.
I assume that the figure at line 406 (pg 9) is the same one as on pg 11 (figure 5). Please correct.
Please define m.m.v.t. before you use the term.
Discussion
Lines 484-496: This paragraph can be made much more concise to state your main findings. You found that examining thermography of the sole before and after exercise was helpful in differentiating the palmar foot pain foot from the other 2 groups. Don’t restate the results (lines 489-496), just highlight your key findings – significantly larger (?) increase in toe area temperature in palmar foot pain compared to other groups and smaller increase in frog temp.
Line 500: The thermography exam does not “confirm” a toe first landing. It may support it.
Lines 500-503: The end of this sentence does not make sense “…as a result of supporting this precisely….” Please reword.
Lines 529-532: Why do you think this happens?
Lines 536-546: It does your reader no good to just report these AUC values. Please recommend what thermography variables/regions of the foot are useful based on this.
Lines 550-569: How does this relate to your study? Please make a connection to your study or delete.
Lines 570-572: You did not measure biomechanics or load in your study so you cannot state this. You can suggest that an increase in toe temperature may be related to differences in loading characteristics of the foot. This could be an area of future study.
Lines 588-592: Please define “other groups”. You could consider something like “non-lame” limbs.
Author Response
Dear reviewer,
Thank you very much for your comments, for your time and effort that helps us to improve the quality of the manuscript. We reorganize the thermographic scan section and give answers in a point-by-point shape to your comments.

This manuscript is a resubmission of an earlier submission. The following is a list of the peer review reports and author responses from that submission.
Round 1
Reviewer 1 Report
Overall, the idea of using thermography to assess horse foot health by measuring surface temperature before and after training is practical and feasible. However, this study has some major issues.
1) sample size is too small to draw any solid conclusions
2) With only providing solo results on temperature changes before and after training, the manuscript is weak in general. More in-depth analysis will be highly needed.
3) The method applied to manually obtain surface temperature using FLIR software tool brings errors in selecting target regions/areas, judging from the photos listed in the manuscript.
Author Response
Dear reviewer,
Thank you for attention to read the manuscript.
In horses with navicular syndrome, according to Turner, the local temperature of the caudal part of the hoof do not have a typical increase in temperature in the palmar aspect of the hoof skin duw to low blood flow. In horses with palmar foot pain there are no information regarding the evolution of thermography in frog and toe area after training. So, this was the reason that we used thermography for evaluation the local temperature of the sole in horses with unilateral palmar foot pain.
Answers:
- We used the obtained information as a pilot study
- Our observation going for evolution of the temperature before and after training in the frog area. We also include information for toe area and we want to observe the differences in thermal pattern of the sole between three groups of limbs (unilateral palmar foot pain limb, contralateral non-lame limb of the affected one and healthy limb from non-lame horses)
- The frog area was named El1, located in the middle third of the frog between the para-cuneal grooves at a size of 8x4 pixels. The toe area was named El2 located between the apex of the frog and hoof wall at a size 6x4 pixels. The soft will detect the minimum and the mean of the temperature inside the circle.
Reviewer 2 Report
This is a very interesting study. However, there are a large number of missing pieces in its current form. Substantial revision is needed so that the useful information can be found by the reader.
Abstract:
Line 31: The first sentence of the methods section is not a complete sentence. Please reword.
In the methods section, the study group is defined, but the control group is not. Please include what horses were included in the control group. How many horses were in each group? What do you mean by “hoof test”? Is this hoof testers? Was the toe surface referring to the solar surface or the hoof wall? Please clarify.
Were statistics performed (what comparisons between feet were done – between limb pairs? Controls vs study horses? Pre- and post-exercise?) Please include a statement about this in the abstract. Creation of ROC curves needs to be included in the methods section.
Line 36: The first sentence of the results section is confusing. Dividing it into 2 sentences may help clarify your main points.
Line 39: What are the 3 studied groups? Earlier controls and study horses were the only 2 groups. What is the 3rd group?
Introduction
In the first paragraph, it may also be worth including the navicular bursitis in the list of conditions with palmar pain.
Lines 52-54: These conditions should be moved up to the first sentence.
Lines 56-59: Replace “is” with “are”.
Lines 63-67: While all of this information is true about MRI, I am not sure that it is necessary for the introduction, as it does not seem to be a key part of this study.
Line 69: Thermography assesses temperature changes, not vascular changes. You could have a vascular change that does not result in temperature change and a temperature change that did not change the vasculature. There is the potential for vascular changes (the word “potential” could be added).
Line 70: Inflammation can result in a temperature change, but cutting off the vascular could also change the temperature. Please reword.
Lines 74-75: Please further define the “variations of the color pattern”. I assume you are talking about the thermography image, not the color pattern of the horse.
Lines 77-79: Please reference these “many studies”.
Lines 84-90: This paragraph seems more suited for the discussion (does not provide critical information for why this study was performed).
Lines 91-100: What is the gap in knowledge that this study is wanting to fill. Please include a statement discussing what the gap of knowledge is.
The objectives/hypotheses need to be well defined. Was it the goal to examine the thermographic pattern of the entire foot pre and post-exercise? To just look at specific regions of the foot pre- and post-exercise? Please clarify and make sure this fits with what is presented in the results section.
Materials and Methods
Lines 106-119: Please separate the study horses out from the controls horses (including signalment, source (all 12 were clinical cases?), shod vs trimmed.
Lines 117-119: I assume that the sample unit was feet (study group). Were only front feet examined? Were both feet from each control horse used in that group? It would be good to include how many feet were included in each group (n= …)
Line 121: What was defined as “positive reaction to hoof testers”? Was this over the heels/across the frog? Please clarify.
Lines 123-124: There are a number of abnormalities of the navicular bone that could categorize a horse as “navicular”. Please include a reference.
Line 125: Was a subjective lameness grading scale used? Please include if that is the case.
Lines 126-127: What is meant by the hoof tester was applied to the frog area?
Line 133: Please replace “pain” with “lameness”.
Line 136: Please include what size needle was used. Was the skin prepared in any way prior to injection (i.e., wiped with alcohol, scrubbed).
Lines 141-143: Please include an approximate angle that was used for these oblique projections.
Lines 145-151: At which gaits were the horses worked (i.e., x min walk, x min trot, x min canter)? Did they all do the same intensity of exercise?
How big was the region that was evaluated over the frog and toe? Was this the same size in each horse? Between horses?
Lines 155-158: Were ambient temperatures and humidity recorded for each horse on each day? I assume the horses were not all examined on the same date.
Line 169: Was the data examined for normality? Is that why the Kruskal-Wallis test was chosen?
Line 175: Is the “navicular bone projection area” the same as the frog? Please keep the terminology consistent throughout.
Lines 180-182: An ROC curve is created from the sensitivity and specificity data and allows creation of cut-off values. Please reword this sentence.
Lines 182-188: The information on how an ROC curve is good for the discussion – it does not need to be included in the statistical analysis section.
Results:
Please include sub-headings in the results. For example: Lameness, Thermography. In general, it is hard to find the key information within the results. It would be helpful to present the toe and frog thermography findings in different paragraphs and highlight when differences were identified (what groups? Before vs after training?). Did the minimum and mean values provide the same results?
The lameness information should be more concise as this is not the main outcome that is being examined. Based on the methods, the study horses were unilaterally lame so this does not need to be stated again: just include how many left and right limbs. The lameness grading scheme should be in the materials section. A median lameness score and range can be included. All of the other information (blocking, hoof testers, frog test) were inclusion criteria, so they should not be included in the results.
Line 218: Why did the control horses present to the hospital since they were not lame? Please include this information.
Line 219: The pastern joint is not the appropriate anatomic term and this joint cannot be flexed on its own. This should be reworded to state distal limb flexion.
Lines 224-232: Were statistical analyses performed to assess regions that appeared warmer or cooler? Did all the feet have this same thermographic appearance or was there some variability?
Line 225: What area is meant by “caudal part of hoof”? Is this the frog region?
Lines 226-232: If you are to report differences in more than the frog and toe regions post training, this should be included earlier in the manuscript (i.e., objectives, methods).
Tables: The information within the tables should be combined to show median and ranges for each group (instead of listing the specimen). This could then be included in a single table.
Figures 3,4, and 5 should be combined into a single figure so it is easier for the reader to see the thermography changes in the 3 groups. Are these from a single representative horse or are all feet combined to form these images?
Lines 260-265: The statistical results need to be related to their comparisons (and should be included in those paragraphs not as a paragraph about Kruskal-Wallis tests). P-values should be presented not the 2 values.
Lines 266-272: Delete this paragraph. You should not need to explain the ranking. This can be reworded to state what was different among the groups (or were some of the groups not significantly different).
Lines 302-304: This paragraph presents results that did not seem to be in the objectives/hypotheses. Please delete or revise objectives/hypotheses (as well as methods) to include this.
Figures 6 and 7: Are there significant differences in any of these groups? If so, please indicate on these graphs.
Discussion
The discussion needs significant work/revision. The beginning of the discussion should highlight main findings of the study. Did you prove/disprove the hypotheses? What do your findings mean? There is a significant amount of redundant material in the first 7 paragraphs. Limitations of thermography should be included but this should be concise and near the end of the discussion.
Lines 387-390: When discussing the ROC curves, the useful variables need to be included. Please add this information.
Lines 399-407: I assume that this is future direction. Please clarify.
Author Response
Dear reviewer,
Thank you for your attention and comments. Also, we appreciate your consideration that this study is interesting. We corrected paper “Evaluation of Thermal Changes of the Sole Surface in Horses with Palmar Foot Pain: A Pilot Study” based on your opinions and comments.
Answers
Abstract
- Line 31: The first sentence of the methods section is not a complete sentence. Please reword.
Answer: Line 33-36. Was composed two study groups represented by eight horses with forelimb unilateral palmar foot pain where the affected limb (n=8) and the contralateral limb (n=8) are separates groups, and a number of four non-lame horses where a unilateral forelimb (n=4) represent the control group.
- Line 36: The first sentence of the results section is confusing. Dividing it into 2 sentences may help clarify your main points.
Answer: Lines 44-47. After training, there were differences in thermal pattern between the sole surface of the affected limb with palmar foot pain from the study group and the others nonlame limbs from the study group and control group. Also, the temperature of the two selected area taken into account present differences between three groups of horse. The temperature of the frog area did not increase in values after training for the affected limb with palmar foot pain.
- Line 39: What are the 3 studied groups? Earlier controls and study horses were the only 2 groups. What is the 3rd group?
Answer: Line 33-36. Was composed two study groups represented by eight horses with forelimb unilateral palmar foot pain where the affected limb (n=8) and the contralateral limb (n=8) are separates groups, and a number of four non-lame horses where a unilateral forelimb (n=4) represent the control group.
Introduction
- Lines 52-54: These conditions should be moved up to the first sentence.
Answer: Lines 59-63: We add navicular bursitis and distal sesamoidean impair ligament to the list of conditions of palmar foot pain.
- Lines 56-59: Replace “is” with “are”.
Answer: Line 67: make the change: Diagnosis of palmar foot pain are based on history…
- Lines 63-67: While all of this information is true about MRI, I am not sure that it is necessary for the introduction, as it does not seem to be a key part of this study.
Answer: we remove the informations about MRI
- Line 69: Thermography assesses temperature changes, not vascular changes. You could have a vascular change that does not result in temperature change and a temperature change that did not change the vasculature. There is the potential for vascular changes (the word “potential” could be added).
Answer: lines 75-76: Thermography is a non-invasive diagnostic method that scans the surface temperature of the animal to find areas that present potential vascular changes.
- Line 70: Inflammation can result in a temperature change, but cutting off the vascular could also change the temperature. Please reword.
Answer: Line 76: This ability to noninvasively assess vascular change makes thermography an ideal imaging tool to aid in the diagnosis….
- Lines 74-75: Please further define the “variations of the color pattern”. I assume you are talking about the thermography image, not the color pattern of the horse.
Answer: Lines 81-84: For thermographyc image, the variation of the color pattern reflect thermal gradients; thus, the warmest areas with increased blood circulation are depicted as being white or red, while the coolest regions with insufficient blood supply appear blue or black.
- Lines 77-79: Please reference these “many studies”.
Answer: Line 87: …. bone lesions [12, 14, 15].
- Lines 84-90: This paragraph seems more suited for the discussion (does not provide critical information for why this study was performed).
Answer: move to conclusion lines 399-405
- Lines 91-100: What is the gap in knowledge that this study is wanting to fill. Please include a statement discussing what the gap of knowledge is.
Answer: Lines 94-109: In horses with navicular syndrome, according to Turner, the local temperature of the caudal part of the hoof do not have a typical increase in temperature in the palmar aspect of the hoof skin duw to low blood flow [23]. In horses with palmar foot pain there are no information regarding the evolution of thermography in frog and toe area after training. We also sought to evaluate the post-training local temperature of the frog area and sole surface of the wall region in horses with palmar foot pain compared to non-lame ones. We hypothesized that the thermal pattern and the local temperature of the frog area and toe area would be similar with increased in local temperature after training in the control group, study group with palmar foot pain and for the study group with non-lame limb from the affected horses.
Materials and Methods
- Lines 106-119: Please separate the study horses out from the controls horses (including signalment, source (all 12 were clinical cases?), shod vs trimmed.
Answers: Lines 112 -129. We made the separation between the groups with the description of each one.
- Lines 117-119: I assume that the sample unit was feet (study group). Were only front feet examined? Were both feet from each control horse used in that group? It would be good to include how many feet were included in each group (n= …)
Answer: We notice the items of each group. 114-117: Two study groups were formed taking into account the affected forelimbs, respectively a study group of the affected palmar foot pain limb (n=8) and a study group of the non-lame contralateral forelimbs (n=8).
Line 126-129: A control group made up of four barefoot non-lame horses was also included in the study with forelimb taken into account (n=4).
- Line 121: What was defined as “positive reaction to hoof testers”? Was this over the heels/across the frog? Please clarify.
Answer: Lines 147. The hoof tester was applied on the frog area by squeeze with one arm the frog and with the other arm the hoof wall to put pressure on the navicular apparatus.
- Lines 123-124: There are a number of abnormalities of the navicular bone that could categorize a horse as “navicular”. Please include a reference.
Answer: Line 162. Add bibliographic sources……. radiographic observation such as enthesophytes on the medial and lateral aspects of the navicular bone, intraosseous cysts and radiolucent or increased opacity of the navicular bone [25, 26].
- Line 125: Was a subjective lameness grading scale used? Please include if that is the case.
Answer: Line 139: Was a subjective lameness score scale takes account of the fact that a horse may appear lamer at the walk than at the trot ( 0 is sound, 10 - non-weight-bearing lameness).
- Lines 126-127: What is meant by the hoof tester was applied to the frog area?
Answer: Lines 147-148. The hoof tester was applied on the frog area by squeeze with one arm the frog and with the other arm the hoof wall to put pressure on the navicular apparatus.
- Line 133: Please replace “pain” with “lameness”.
Answer: Line 152, made the change. To localize the lameness.
- Line 136: Please include what size needle was used. Was the skin prepared in any way prior to injection (i.e., wiped with alcohol, scrubbed).
Answer: Line 156: Perineural analgesia of the medial and lateral digital palmar nerve was performed using 1.5-mililiter solution of 2% mepivacaine (Scandicaine®) for each nerve and the size of the neddle was 25 gauge and 16 mm [24].
- Lines 141-143: Please include an approximate angle that was used for these oblique projections.
Answer: Lines 161-162: …..65°dorsoproximal–palmarodistal oblique (DPr-PaDiO) and 45°palmaroproximal–palmarodistal oblique (PaPr-PaDIO).
- Lines 145-151: At which gaits were the horses worked (i.e., x min walk, x min trot, x min canter)? Did they all do the same intensity of exercise?
Answer: Lines 170: All the horses were subjected for the same intensity of the exercise: 15 minutes walk on left and right hand and 15 minutes trot on left and right hand.
- How big was the region that was evaluated over the frog and toe? Was this the same size in each horse? Between horses?
Answer: Lines 172: El1 frog area between the para-cuneal grooves at a size of 8x4. The toe area was named El2 located between the apex of the frog and hoof wall at a size 6x4 pixels.
- Lines 155-158: Were ambient temperatures and humidity recorded for each horse on each day? I assume the horses were not all examined on the same date.
Answer: Line 182-183: All measurements for each horse of the study group were conducted during the same season in the same environmental conditions, i.e., dry sand, air temperature of 18–22 °C, humidity between 60% and 70%, without extreme heat or cold weather. Line: 195-196: For horses of the control group the thermographic evaluation was performed in the same day.
- Line 169: Was the data examined for normality? Is that why the Kruskal-Wallis test was chosen?
Answer: Line 198-199: For this, the Kruskal–Wallis statistical test was used for normally distribution data and was applied to the temperature, and four procedures were used to evaluate each group.
- Line 175: Is the “navicular bone projection area” the same as the frog? Please keep the terminology consistent throughout.
Answer: Line 200: Yes, was an error, we change it ….temperatures of the frog area before…
- Lines 180-182: An ROC curve is created from the sensitivity and specificity data and allows creation of cut-off values. Please reword this sentence.
Answer: line 205-208: The Receiver Operating Characteristic Curve (ROC curve) is created from the sensitivity and specificity data and allows creation of cut-off values. The purpose of the ROC analysis is to identify the optimal value to differentiate a positive from a negative result by obtain the value area under the curve (AUC).
- Lines 182-188: The information on how an ROC curve is good for the discussion – it does not need to be included in the statistical analysis section.
Answer: this part was moved to discussion at line 427-431.
Results:
Please include sub-headings in the results. For example: Lameness, Thermography. In general, it is hard to find the key information within the results. It would be helpful to present the toe and frog thermography findings in different paragraphs and highlight when differences were identified (what groups? Before vs after training?). Did the minimum and mean values provide the same results?
Answer: We separate the result in two chapter 3.1. Lameness exam and 3.2. Thermographic scan. For each group we present individual the results obtained for frog and toe area.
- The lameness information should be more concise as this is not the main outcome that is being examined. Based on the methods, the study horses were unilaterally lame so this does not need to be stated again: just include how many left and right limbs. The lameness grading scheme should be in the materials section. A median lameness score and range can be included. All of the other information (blocking, hoof testers, frog test) were inclusion criteria, so they should not be included in the results.
Answer: We add only the result obtained for each test in this section. We check the response to perineural analgesia after 30 minutes and a graphic representation of the lameness score was included.
- Line 218: Why did the control horses present to the hospital since they were not lame? Please include this information.
Answer: line 128-130: These horses were brought to clinic to participate in the study with the consent of the owners without presenting a musculoskeletal condition.
- Line 219: The pastern joint is not the appropriate anatomic term and this joint cannot be flexed on its own. This should be reworded to state distal limb flexion.
Answer: Line 238. Change with “distal limb flexion”.
- Lines 224-232: Were statistical analyses performed to assess regions that appeared warmer or cooler? Did all the feet have this same thermographic appearance or was there some variability?
Answer: line 198: The statistical analyses was performed to check the minimum and the mean temperature for the two area taken into account before and after training. Line 306-321: There were differences of the thermographic appearance between the groups.
- Line 225: What area is meant by “caudal part of hoof”? Is this the frog region?
Answer: Line 265: frog area.
- Lines 226-232: If you are to report differences in more than the frog and toe regions post training, this should be included earlier in the manuscript (i.e., objectives, methods).
Answer: 103-105: We also sought to evaluate the post-training local temperature of the frog and sole region and the thermal pattern in horses with palmar foot pain compared to nonlame ones.
- Tables: The information within the tables should be combined to show median and ranges for each group (instead of listing the specimen). This could then be included in a single table
Answer: Line 272-306: insert a table with median and ranges for each group. The previous table are supplementary materials.
- Figures 3,4, and 5 should be combined into a single figure so it is easier for the reader to see the thermography changes in the 3 groups. Are these from a single representative horse or are all feet combined to form these images?
Answer: Line 306-321. we add all the images obtained before and after training for each groups one after another in the manuscript. For each group is the image obtained for the same horse before and after training.
- Lines 260-265: The statistical results need to be related to their comparisons (and should be included in those paragraphs not as a paragraph about Kruskal-Wallis tests). P-values should be presented not the 2
Answer: Line 322-342: we add only the p-value in the manuscript.
- Lines 266-272: Delete this paragraph. You should not need to explain the ranking. This can be reworded to state what was different among the groups (or were some of the groups not significantly different).
Answer: we delete the paragraph.
- Lines 302-304: This paragraph presents results that did not seem to be in the objectives/hypotheses. Please delete or revise objectives/hypotheses (as well as methods) to include this.
Answer: Answer: 103-105: We complete the objectives and hypothesis with information about toe, frog area and thermal pattern of the sole in horses with palmar foot pain before and after training.
- Figures 6 and 7: Are there significant differences in any of these groups? If so, please indicate on these graphs.
Answer: Lines 346-354: we mark the relevant changes on the graphics.
Discussion
- The discussion needs significant work/revision. The beginning of the discussion should highlight main findings of the study. Did you prove/disprove the hypotheses? What do your findings mean? There is a significant amount of redundant material in the first 7 paragraphs. Limitations of thermography should be included but this should be concise and near the end of the discussion.
Answer: We start the discussion chapter with information about the hypothesis and after with information about the changes that appeared for each group and areas taken into account. We made changes and cut pat of the information that was in the discussion section. The limitation of the study we add in the final of the manuscript.
- Lines 387-390: When discussing the ROC curves, the useful variables need to be included. Please add this information.
Answer: Lines 432-436. Add information about variable. maximum value for AUC is 1 and indicates a perfect diagnostic test able to correctly identify both disease and those healthy subjects. If the obtained AUC is greater than 0.9, then the result is excellent. If the obtained area is between 0.8 and 0.9 the result is very good, if it is between 0.7 and 0.8 the result is good, if it is between 0.6 and 0.7 the result is fair, and if it is less than 0.6 the result is dismissed.
- Lines 399-407: I assume that this is future direction. Please clarify
Answer: Further investigation: we propose to use a soft to measure the surface of the temperature using 3D Profilometry. Another direction to check if there exist changes in local temperature and thermal pattern of the sole between the horses with unilateral palmar foot pain and those with bilateral or subclinical palmar foot pain.
Reviewer 3 Report
While the concept and study design are interesting, this manuscript is poorly written and does not present the materials and methods, results, and discussion in a manner that is easy to read or easy to understand. The authors’ gathered a great deal of data prior to the start of the study but failed to describe grading scales or enough detail about data collection to provide a clear picture of what was actually done. The results section was confusing and hard to follow and needs significant reorganization. The first portion of the discussion simply repeats the material and methods and results section rather than providing a true discussion and interpretation of your findings. The reviewer therefore feels that this manuscript needs major revisions prior to publication.
Line by line comments are as follows:
Summary:
Lines 17-18: Reword sentence “…conducted to investigate the accuracy of thermography to detect changes in local temperature in horses with…”
Line 20: What is meant by “toe area surface”? Please clarify here and in rest of manuscript
Line 22: What are the 3 groups of horses? The summary should be able to read so that the reader can understand the basics of how your study was conducted.
Line 25: What is meant by “healthy one”. Please clarify here and throughout the manuscript.
Line 25: Replace “are” with “is” (further investigation IS required)
Line 25-26: This sentence is confusing. Please rephrase
Abstract:
Line 38-39: What was the sensitivity/specificity? What information did you get from the ROC curve?
Line 41: “Is” instead of “are”
Introduction:
Line 47: What collateral ligaments are you referring to? There are multiple in the foot
Line 52: Impar is misspelled
Line 63: “has” instead of “had”
Line 71: Condition and horse should be plural
Line 77-79: Add in references at the end of sentence
Line 87: “Benefitted” is not best word choice; “performed” physical therapy?
Line 88: Anesthetic protocol? What does this mean? Not undergone general anesthesia? Sedation? Please clarify
Line 98-99: Do you mean just navicular region? Please rephrase
Line 101-103: You have not discussed your groups yet, so this sentence is confusing and needs clarification.
Line 108-109: Were healthy horses shod or barefoot?
Line 112-113: How lame were the horses? How was lameness determined and by whom?
Line 116: What made the farrier qualified? Years of experience? Training?
Line 116-117: How long were horses off NSAIDs?
Line 117-119: Need to better define the 3 groups
Line 120: Define chronic. Two months duration? Two years?
Line 120: List exclusion criteria
Line 139: Approximate angle of DP oblique taken?
Line 141-143: Inclusion criteria for control group?
Line 143: Use “perineural analgesia” instead of “nerve block”
Line 166: Was operator blinded to group assignment or unblinded? If unblinded, need to state this and include as limitation in discussion
Line 169-172: You don’t need to include this information
Results:
Please add in horse demographics here. The discussion of when horses became lame is not needed.
Lines 201-203: How was frog test and hooftester grading performed? Just positive or negative? Mild, moderate, marked?
Lines 204-205: In reviewer’s experience, horses rarely become 100% sound after perineural anesthesia. Can you describe the degree of improvement further?
Lines 207-212: Again if you are going to describe radiographs and use them for inclusion, you need more detail as to how they were graded and severity of findings.
Lines 211-212: Increased opacity of the flexor cortex does not typically occur. Thickening of flexor cortex? Sclerosis of medullary cavity?
Line 221: “Hoof tester” is misspelled
Line 222-223: Opacity in articular or flexor borders? Increased, decreased, abnormal? Can you describe these changes further
Tables: Would be better to use tables to show statistics rather than your raw data. Raw data could be included as a supplemental table. Include p-value in tables as well as in text. Need a heading for all tables.
Figure 3: Remove temperature boxes on your images and need more descriptors/arrows pointing out changes.
Line 260-263: Listing the x2 vaue is not needed in text, just p-values for everything you tested. i.e. “There was a significant increase in temperature comparing the non-lame foot pre and post training in the frog area (P < 0.0001)…”
Overall, your results section is confusing and hard to follow. Needs significant reorganization, as it is hard to follow what all you were actually comparing.
Figure 6 and 7: Delimited is not a word
Figure 8: Your legend is not explained. Need much more information and you should not reference supplementation data in your figures in main manuscript.
Discussion:
Lines 328-333: This is just repeating your materials and methods. Please omit and use discussion to briefly describe findings and why you think you got these results. You start to do this in Line 359.
Author Response
Dear reviewer,
Thank you for your attention and comments. We corrected paper “Evaluation of Thermal Changes of the Sole Surface in Horses with Palmar Foot Pain: A Pilot Study” based on your opinions and comments.
Summary
- Lines 17-18: Reword sentence “…conducted to investigate the accuracy of thermography to detect changes in local temperature in horses with…”
Answer: Lines 17-18, 31-33: A pilot study was conducted to investigate the accuracy of thermography to detect changes in local temperature and to compare the thermal patterns observed on the sole surface after training.
- Line 20: What is meant by “toe area surface”? Please clarify here and in rest of manuscript
Answer: Line 170. The toe area was named El2 located between the apex of the frog and hoof wall…
- Line 22: What are the 3 groups of horses? The summary should be able to read so that the reader can understand the basics of how your study was conducted.
Answer: Lines 33-36: Was composed two study groups represented by eight horses with forelimb unilateral palmar foot pain where the affected limb (n=8) and the contralateral limb (n=8) are separates groups, and a number of four non-lame horses where a unilateral forelimb (n=4) represent the control group.
- Line 25: What is meant by “healthy one”. Please clarify here and throughout the manuscript.
Answer: lines 35-36: represent the control group of the non-lame horses.
- Line 25: Replace “are” with “is” (further investigation IS required)
Answer: line 28: change in text
- Line 25-26: This sentence is confusing. Please rephrase
Answer: 27-28: Our results indicate that thermography can detect changes in local temperature of the sole surface for horses with palmar foot pain and non-lame ones after training.
Abstract
- Line 38-39: What was the sensitivity/specificity? What information did you get from the ROC curve?
Answer: Lines 424-428: For specificity/sensitivity: AUC of 1 and p<0.05, shown a capable diagnostic test to correctly discriminate between non-lame horses and those with palmar foot pain. One variable had a very good result, with an AUC of 0.8 and p<0.05. For one variable, represent the mean temperature of toe area after training, the result was fair with an AUC of 0.6 and p>0.05 (p=0.217) with worthless diagnostic accuracy.
Lines 429-434: ROC curve confirm the medical use of thermography with an excellent results for six of the eight examined variables, with an AUC of 1 and p<0.05 and shown a capable diagnostic test to correctly discriminate between the non-lame horses and those with palmar foot pain.
- Line 41: “Is” instead of “are”
Answer: Line 54: correct with “is” Further investigation is..
Introduction:
- Line 47: What collateral ligaments are you referring to? There are multiple in the foot
Answer: line 61: complete with ..desmitis of the collateral ligaments of the distal interphalangeal joint (DIP joint)…
- Line 52: Impar is misspelled
Answer: Line 60: correct with distal sesamoidean impar ligament
- Line 63: “has” instead of “had”
Answer: we gave up to this paragraph about MRI
- Line 71: Condition and horse should be plural
Answer: Line 78, correct with … the diagnosis of certain lameness conditions in horses .
- Line 77-79: Add in references at the end of sentence
Answer: line 87: we add the references in the end … bone lesions [12, 14, 15].
- Line 87: “Benefitted” is not best word choice; “performed” physical therapy?
Answer: correct with “performed” at Line 399: ….not have performed physical therapy.
- Line 88: Anesthetic protocol? What does this mean? Not undergone general anesthesia? Sedation? Please clarify
Answer: We don’t administered any sedation to horses before thermography, only perineural analgesia for the digital palmar nerve. The thermographic examination was performed the next day after clinical exam and perineural analgesia.
- Line 98-99: Do you mean just navicular region? Please rephrase
Answer: Lines 103-105: We proposed to evaluate the post-training local temperature evolution of the frog and sole region and the thermal pattern between horses with unilateral palmar foot pain and the contralateral non-lame one and non-lame horses.
- Line 101-103: You have not discussed your groups yet, so this sentence is confusing and needs clarification.
Answer: Lines 33-36: Was composed two study groups represented by eight horses with forelimb unilateral palmar foot pain where the affected limb (n=8) and the contralateral limb (n=8) are separates groups, and a number of four non-lame horses where a unilateral forelimb (n=4) represent the control group.
- Line 108-109: Were healthy horses shod or barefoot?
Answer: Line 126: The horses from the control group were barefoot.
- Line 112-113: How lame were the horses? How was lameness determined and by whom?
Answer:
Lines 138-139: The examination was performed by two veterinarians.
Lines 212-216: Three horses were lame on the left forelimb and five were lame on the right forelimb. All horses had a positive response to pain after flexion of the distal forelimb, which produced a transient exacerbation of the lameness. A lameness score of four was found in two of the studied horses, a score of six was found in three horses and seven in three horses, considering the European scale for lameness.
- Line 116: What made the farrier qualified? Years of experience? Training?
Line 124: We choose the farrier as a result of his experiences in horseshoe of the race horses.
- Line 116-117: How long were horses off NSAIDs?
Answer: Lines 125-126: No anti-inflammatory medication was administrated to the horses in the last three weeks before the examination
- Line 117-119: Need to better define the 3 groups
Answer: Lines 33-36: Was composed two study groups represented by eight horses with forelimb unilateral palmar foot pain where the affected limb (n=8) and the contralateral limb (n=8) are separates groups, and a number of four non-lame horses where a unilateral forelimb (n=4) represent the control group.
- Line 120: Define chronic. Two months duration? Two years?
Answer: Line 131. …more than three months of progressive unilateral forelimb lameness.
- Line 120: List exclusion criteria.
Answer: Lines 136-139: The exclusion criteria for the study group was represented by superficial and deep flexors tendinitis, arthritis/arthrosis of the metacarpo-phalangeal joint and other musculoskeletal injuries that include the acropodial region.
- Line 139: Approximate angle of DP oblique taken?
Answer: Lines 160 – 162: …..65°dorsoproximal–palmarodistal oblique (DPr-PaDiO) and 45°palmaroproximal–palmarodistal oblique (PaPr-PaDIO).
- Line 141-143: Inclusion criteria for control group?
Answer: Lines 241-246: The horses from the control group did not present any history of lameness and demonstrated no pain reaction after distal limb flexion. The clinical examination revealed no sign of lameness after walking, trotting and lunging. The horses did not present any reaction to pain after applying the hoof tester and frog wedge test. The radiographic images did not reveal any changes in the structure of the navicular bone or abnormal opacity in the articular and flexor borders.
- Line 143: Use “perineural analgesia” instead of “nerve block”
Answer: Lines 164-165: For the control group, the lateral and medial digital nerve did not receive a perineural analgesia.
- Line 166: Was operator blinded to group assignment or unblinded? If unblinded, need to state this and include as limitation in discussion
Answer: Line 449: The operator was unblended during the marking of the study areas
- Line 169-172: You don’t need to include this information
Answer: cut of this paragraph.
Results:
We separate the results in two sections: Lameness exam and Thermography scan
In the lameness exam present the results obtain after the clinical examination.
- Lines 201-203: How was frog test and hooftester grading performed? Just positive or negative? Mild, moderate, marked?
Answer: Line 147-151: The hooftester was applied on the frog area by squeeze with one arm the frog and with the other arm the hoof wall to put pressure on the navicular apparatus. The frog wedge test was performed by applying a wooden pad that has 10 centimeters long and 5 centimeters wide with an angle of 15 degree under the palmar two thirds of the frog and forcing the horse to stand on that foot for 1 minute by lifting the contralateral limb. Line 220-223: Six of the horses presented a mild reaction and two horses had a moderate reaction of retraction after applying the hooftester on the frog area of the hoof. After using the wedge frog test three of the horses present a mild reaction, three horses a moderate reaction and two horses an obvious reaction.
- Lines 204-205: In reviewer’s experience, horses rarely become 100% sound after perineural anesthesia. Can you describe the degree of improvement further?
Answer: Lines 224-229: After perineural analgesia of the medial and lateral digital nerve, the gait improved in one horse from four score to soundness, in two horses from score seven, respectively six to soundness, in three horses the lameness score reduce at one and in two horse the lameness score reduced at two. After perineural analgesia, the response to hooftester on 228 the palmar part of the hoof was negative in six horses and two horses present a mild reaction.
- Lines 207-212: Again if you are going to describe radiographs and use them for inclusion, you need more detail as to how they were graded and severity of findings.
Answer: Lines 230-237. Complete with information 65°DPr-PaDiO radiographs of the hooves were taken for each of the horses included in the study group, revealing enthesophytes on the medial and lateral aspects of the navicular bone in three of the horses, along with sclerosis of medullary cavity in two horses and bone cysts in the navicular bone two horses (Figure 2), while PaPr-PaDIO radiographs revealed intraosseous cysts in two and increased opacity on the flexor border of the navicular bone for three horses. The LM revealed increased opacity of the flexor cortex of the navicular bone for two horses. Multiple lesions was observe in case of six horse and only in two horses was observe an singular lesion.
- Lines 211-212: Increased opacity of the flexor cortex does not typically occur. Thickening of flexor cortex? Sclerosis of medullary cavity?
Answer: Lines 232, 236: Was observe sclerosis of the medullary cavity and the LM reveal ncreased opacity of the flexor cortex of the navicular bone for two horses.
- Line 221: “Hoof tester” is misspelled
Answer: Line 244: correct with hooftester.
Line 222-223: Opacity in articular or flexor borders? Increased, decreased, abnormal? Can you describe these changes further
Answer: 240 -244: The radiographic images did not reveal any changes in the structure of the navicular bone or abnormal opacity in the articular and flexor borders in the LM view. The 65°DPr-PaDiO radiographs of the hooves didn’t show any enthesophytes or arthosis of the DIP and the PaPr-PaDIO radiographs not detect a bone cyst in the navicular bone.
- Tables: Would be better to use tables to show statistics rather than your raw data. Raw data could be included as a supplemental table. Include p-value in tables as well as in text. Need a heading for all tables.
Answer: We add table with statistical data and the raw data we included as a supplementary. A table with p value is add on the supplementary data.
- Figure 3: Remove temperature boxes on your images and need more descriptors/arrows pointing out changes.
Answer: we didn’t succeed in all the images to remove the temperature box but we add more arrow and include a better descripton for each image.
- Line 260-263: Listing the x2 vaue is not needed in text, just p-values for everything you tested. i.e. “There was a significant increase in temperature comparing the non-lame foot pre and post training in the frog area (P < 0.0001)…”
Answer: Line 322: We remove the X2 value and change the desctiption with There was a significant increase in temperature comparing the three groups before 323 and after training in the frog area (p < 0.0001).
- Overall, your results section is confusing and hard to follow. Needs significant reorganization, as it is hard to follow what all you were actually comparing.
Answers: Your comments help us to show a better presentation. Hope this new version of the results to be clear.
Discussion:
- Lines 328-333: This is just repeating your materials and methods. Please omit and use discussion to briefly describe findings and why you think you got these results. You start to do this in Line 359.
Answer: We remove the beginning part of the conclusion and reorganize all of this part. We hope the new version of the conclusion to be relevant.